# Massively parallel profiling of RNA-targeting CRISPR-Cas13d

Hung-Che Kuo[1] ✉, Joshua Prupes[1], Chia-Wei Chou [1] & Ilya J. Finkelstein [1,2] ✉

CRISPR-Cas13d cleaves RNA and is used in vivo and for diagnostics. However, a systematic understanding of its RNA binding and cleavage specificity is lacking. Here, we describe an RNA Chip-Hybridized Association-Mapping Platform (RNA-CHAMP) for measuring the binding affinity for > 10,000 RNAs containing structural perturbations and other alterations relative to the CRISPR RNA (crRNA). Deep profiling of Cas13d reveals that it does not require a protospacer flanking sequence but is exquisitely sensitive to secondary structure within the target RNA. Cas13d binding is penalized by mismatches in the distal crRNA-target RNA region, while alterations in the proximal region inhibit nuclease activity. A biophysical model built from these data reveals that target recognition initiates in the distal end of the target RNA. Using this model, we design crRNAs that can differentiate between SARS-CoV-2 variants by modulating nuclease activation. This work describes the key determinants of RNA targeting by a type VI CRISPR enzyme.

Class 2 CRISPR-Cas systems are useful for genetic engineering because they target DNA and/or RNA with a single effector protein[1]. Among class 2 enzymes, Cas13 subtypes exclusively target and cleave RNA[2–9]. Cas13s process their CRISPR-RNAs (crRNAs), bind a target RNA that is complementary to the crRNA, and cleave the target RNA (*cis*-cleavage) and other RNA molecules via a non-specific RNase activity (*trans*-cleavage)[10,11]. These RNase activities are catalyzed by two Cas13-encoded higher eukaryotes and prokaryotes nucleotide-binding (HEPN) domains which can be mutagenically inactivated to convert Cas13 into an RNA-binding module[10,12–14]. Due to these activities, Cas13 variants are broadly used in vitro and in cells[5,15–17]. For example, Cas13d —one of the most compact and biochemically active Cas13 enzymes— can efficiently knockdown RNA in mammalian cells and animal models[5,18–25]. Moreover, Cas13d fusions are used for RNA tracking, editing, modification, and splicing regulation[5,7,26,27]. Cas13d has also been applied for nucleic acid detection in CRISPR diagnostics[28,29]. However, the binding and cleavage specificity of Cas13d on partially matched target RNAs has not been fully characterized, limiting our understanding and biotechnological applications of this enzyme.

Biochemical studies have reported various targeting specificities across Cas13-family enzymes. Some enzymes require a protospacer flanking sequence (PFS)—a specific sequence adjacent to the target— for RNA cleavage. For example, LshCas13a prefers a non-G 3′-PFS, whereas BzCas13b favors non-C 5′-PFS and 3′PFS of NNA or NAN[2,3,6]. However, LwaCas13a, PspCas13b, EsCas13d, RfxCas13d (CasRx), and Cas13X.1 may not require any PFS at all[2,4,5,7,13,16,17]. The cleavage activity of LwaCas13a, LshCas13a, and LbuCas13a is sensitive to mismatches in the central region of crRNA-target RNA duplex[2,16,30,31]. Large-scale Cas13d screens in mammalian cells also concluded that Cas13d is largely intolerant to mismatches in the distal spacer region (positions 15-21)[32,33]. Additionally, prior reports suggested that the secondary structure of the target is negatively correlated with Cas13d targeting efficiency[4,32,34]. These experiments primarily use Cas13 cleavage as an output, conflating binding, activation, and cleavage into a single reporter. Interpreting studies across different experimental conditions and target RNAs is especially challenging because RNA structure can change drastically even with a single nucleotide substitution and may also impact both binding and cleavage. A complete understanding of off-target activity requires the biochemical separation of binding and cleavage across a defined set of structural target RNA and sequence perturbations.

Here, we describe RNA-CHAMP (Chip-Hybridized Association-Mapping Platform) for massively parallel profiling of RNA-protein interactions on a conventional microscope and the nearly ubiquitous

[1]Department of Molecular Biosciences and Institute for Cellular and Molecular Biology, University of Texas at Austin, Austin, TX 78712, USA. [2]Center for Systems and Synthetic Biology, University of Texas at Austin, Austin, TX 78712, USA. ✉e-mail: hckuo@utexas.edu; ilya@finkelsteinlab.org

chips that are discarded at the end of Illumina-based sequencing. Our approach differs from prior high-throughput methods[35,36] that repurpose the obsolete Illumina Genome Analyzer IIx instruments and require custom hardware modifications[37]. Using RNA-CHAMP, we characterize how target RNA alterations impact the RNA binding by Cas13d. Contrary to other Cas13-family enzymes, Cas13d does not have a strong PFS preference. However, nucleotide substitutions that increase the overall target RNA secondary structure profoundly decrease the binding affinity. Mismatches and intramolecular base pairing in the distal region of the target RNA strongly decrease Cas13d binding. Surprisingly, mismatches in the proximal region of the target do not affect binding but inhibit nuclease activity. A series of biophysical models of increasing complexity shed insights into the mechanism of Cas13d binding. Together, our results and model suggest that Cas13d initially recognizes the target RNA in the solvent-exposed distal spacer region, followed by RNA duplex formation towards the target RNA in the proximal region. Structural elements in the distal segment impede Cas13d binding. Using these insights, we design a series of partially mismatched crRNAs to detect single nucleotide polymorphisms (SNPs) in circulating SARS-CoV-2 variants. These results will guide future RNA editing and CRISPR diagnostics applications. More broadly, RNA-CHAMP will enable high-throughput mapping of protein-RNA interactions in diverse cellular processes.

## Results

### RNA-CHAMP measures protein-RNA interactions on sequenced Illumina chips

RNA-CHAMP repurposes Illumina next-generation sequencing (NGS) chips to quantify millions of protein-RNA interactions (Fig. 1A). RNA molecules are transcribed in situ from a template DNA library that has been sequenced using an Illumina MiSeq instrument. We designed the DNA library with the T7 RNA polymerase (RNAP) promoter, a variable region of interest, and the RNAP-stalling *TerB* DNA sequence[36,38]. This DNA sequence is recognized by Tus, a bacterial protein that blocks T7 RNAP translocation[39,40]. The identity and physical coordinates of each DNA cluster are determined during NGS. After sequencing, the chip is regenerated to remove leftover fluorescent nucleotides and resynthesize the double-stranded (ds) DNA[41]. Tus is then added to the chip to stall T7 RNAP. In vitro transcription (IVT) and subsequent stalling of T7 RNAP tethers the transcript to its DNA template. Polymerases that stall prematurely can undergo recycling or exchange with an active enzyme[42]. In both scenarios, the transcript is generated after a transcribing RNAP is stalled by Tus.

We first assayed the efficiency of RNA capture on the MiSeq chip. To confirm that Tus recognizes *TerB*-encoding DNA clusters, we purified FLAG-epitope labeled Tus and fluorescently labeled it with an ATTO488-conjugated anti-Flag antibody[43] (Fig. S1A). We sequenced a library that included DNAs with and without the *TerB* sequence. Over 90% of *TerB*-encoding DNA clusters co-localized with fluorescent Tus (Fig. S1A). The remaining *TerB*-encoding clusters could not be resolved by our image processing software, usually due to their spatial overlap. Importantly, Tus did not bind clusters that lacked *TerB*. All downstream analysis was conducted on *TerB*-containing DNA clusters. To confirm that the RNA transcripts are stably retained after IVT, we hybridized a complementary ATTO647N-labeled oligonucleotide to the RNA transcripts in situ (Fig. S1B). The chip also included DNA clusters with scrambled T7 RNAP promoters as negative controls. We observed an RNA signal from ~90% of promoter-containing clusters, but not from scrambled promoter clusters (Fig. S1B). These results demonstrate that RNA-CHAMP can generate libraries of user-defined RNA molecules on repurposed MiSeq chips.

Next, we characterized the specificity and off-target RNA binding of *Eubacterium siraeum* (*Es*) Cas13d, a prototypical member of the CRISPR RNA-guided RNA nucleases[4,5]. We purified nuclease-dead *Es*Cas13d with an N-terminal SNAP-tag and fluorescently labeled it

with SNAP-Surface-488 (hereafter referred to as "dCas13d"; Fig. S1C). The ribonucleoprotein (RNP) complex was reconstituted to 100% homogeneity by incubating dCas13d with a 4-fold excess of the crRNA followed by size exclusion chromatography. Native gel electrophoresis confirmed complete RNP formation (Fig. S1D). This procedure was repeated for RNPs with different crRNAs and used in all subsequent experiments. The SNAP-tag did not alter the protein's RNA-binding affinity, as measured via Biolayer Interferometry (BLI) (Fig. S1E).

Type VI CRISPR-Cas nucleases recognize a protospacer-flanking sequence (PFS) that is immediately adjacent to the 5' or 3' of the target RNA[2–7]. To test whether *Es*Cas13d is sensitive to the PFS, we included three randomized bases on both the 5' and 3' of the matched target sequence. In addition, the target RNA library included up to two mismatches, insertions, or deletions relative to the crRNA (Fig. 1B & Source Data). To confirm that our findings are generalizable across targets, we also prepared a second library with a different target RNA sequence but identical design characteristics (Fig. S3 & S4). We sequenced both RNA libraries to ensure >-10–100 DNA clusters for all library members (Fig. 1B, right). We also included unrelated DNA sequences as controls or fiducial markers for downstream image analysis and spatial registration. After sequencing, the MiSeq chip was regenerated and transcribed with T7 RNAP for downstream experiments.

Transcribed libraries were incubated with increasing concentrations of dCas13d (Fig. 1C, D). Clusters with T7 promoters showed dCas13d concentration-dependent increases in fluorescence intensities, whereas scrambled promoters showed no dCas13d binding (Fig. 1C). The fluorescent intensities of clusters across all concentrations were background-subtracted and fit with a Hill equation without cooperativity to determine the apparent binding affinity (ABA) (Fig. 1D & Methods)[41,44]. To directly compare the relative binding affinity across the entire library, we calculated the change in the binding affinity (ΔABA) as the natural logarithm of the matched target affinity divided by partially matched RNA library members (see Methods). The ΔABA reports the relative change in dCas13d binding affinity of every library member relative to a reference (matched target) sequence. Two biological replicates showed excellent reproducibility across the entire dynamic range of binding affinities (Fig. 1E). In a partially matched library, we measured the binding affinities for 3893 sequences from the target library out of 4936 total members (Fig. 1F). The remaining target RNA sequences had binding affinities or fluorescent signals that were below our detection limit. Using BLI, we validated a subset of 16 RNA targets across the entire dynamic range of the RNA-CHAMP experiments, including sequences with mutations in the target RNA as well as the PFS (Fig. S2). ABAs calculated from BLI measurements were in excellent agreement with the sequences from our library, indicating that RNA-CHAMP accurately captures the relative affinities of dCas13d to its target RNA sequences (Pearson's $r = 0.89$; Fig. 1G, S2). Moreover, the BLI analysis indicates that the ΔABA is dominated by $k_{on}$, likely because the target RNA-crRNA duplex is very stable after hybridization (Table S1). We conclude that the massively parallel RNA-CHAMP platform can quantitatively profile protein-RNA interactions.

### Cas13d requires a partially unstructured target RNA in the distal region

We measured dCas13d binding affinity with a PFS library consisting of three random nucleotides on the 5' and 3' end of the 22 nt matched target sequence (target #1) (Fig. 2). We measured ΔABAs for a total of 1457 PFS combinations. The remaining sequences were below our detection threshold. Although dCas13d exhibited a ~3-fold difference in ΔABAs across the entire PFS dataset, it did not have a strong PFS preference (Fig. 2A). We observed a similar result in a second target (target #2) library but with a slight preference for non-G 3'-PFS (position −1) (Fig. S3). Combining the top 25% highest ABA binding sequences in both targets confirms that Cas13d has a weak preference for the 3'-PFS (Fig. S3D). This weak PFS preference, however, doesn't

explain the broad range of ΔABAs that we measured across the library of matched target RNA sequences.

We reasoned that the target RNA secondary structure can regulate Cas13d binding[4,32,34]. When inspecting both high- and low-affinity target RNAs, we observed that dCas13d prefers target RNAs that are not predicted to be base paired in the distal region (positions 11-22) (Fig. 2A–C & S3A, B)[45]. For example, the 5'-PFS GUA forms a stem with the 5' constant region and exposed positions 19-22, which resulted in ~2-fold stronger dCas13d binding than 5'-PFS UAA (Fig. 2A, B). Similarly, 3'-PFS GCU forms a stem with the 3' constant region and exposed position 14-20. These exposed distal nucleotides in 3'-PFS GCU led to

a ~2-fold increase in dCas13d binding affinity relative to 3'-PFS UGG (Fig. 2A, C). BLI measurements independently validated these observations (Fig. 2D). This also confirms that low-affinity PFSs have a similar off-rate ($k_d$), but slower on-rates ($k_a$) than high-affinity PFSs (Fig. 2D, Table S1). These results highlight that RNA structure regulates Cas13d access to the matched target RNA.

To determine how the local target RNA structure affects Cas13d binding, we computed the number of predicted intramolecular base pairs in the proximal (positions 1-11) and distal (positions 12-22) regions of the target RNA. Intramolecular base pairs can form with the RNA outside the target, or within the target itself. For example, the 3'-GCU

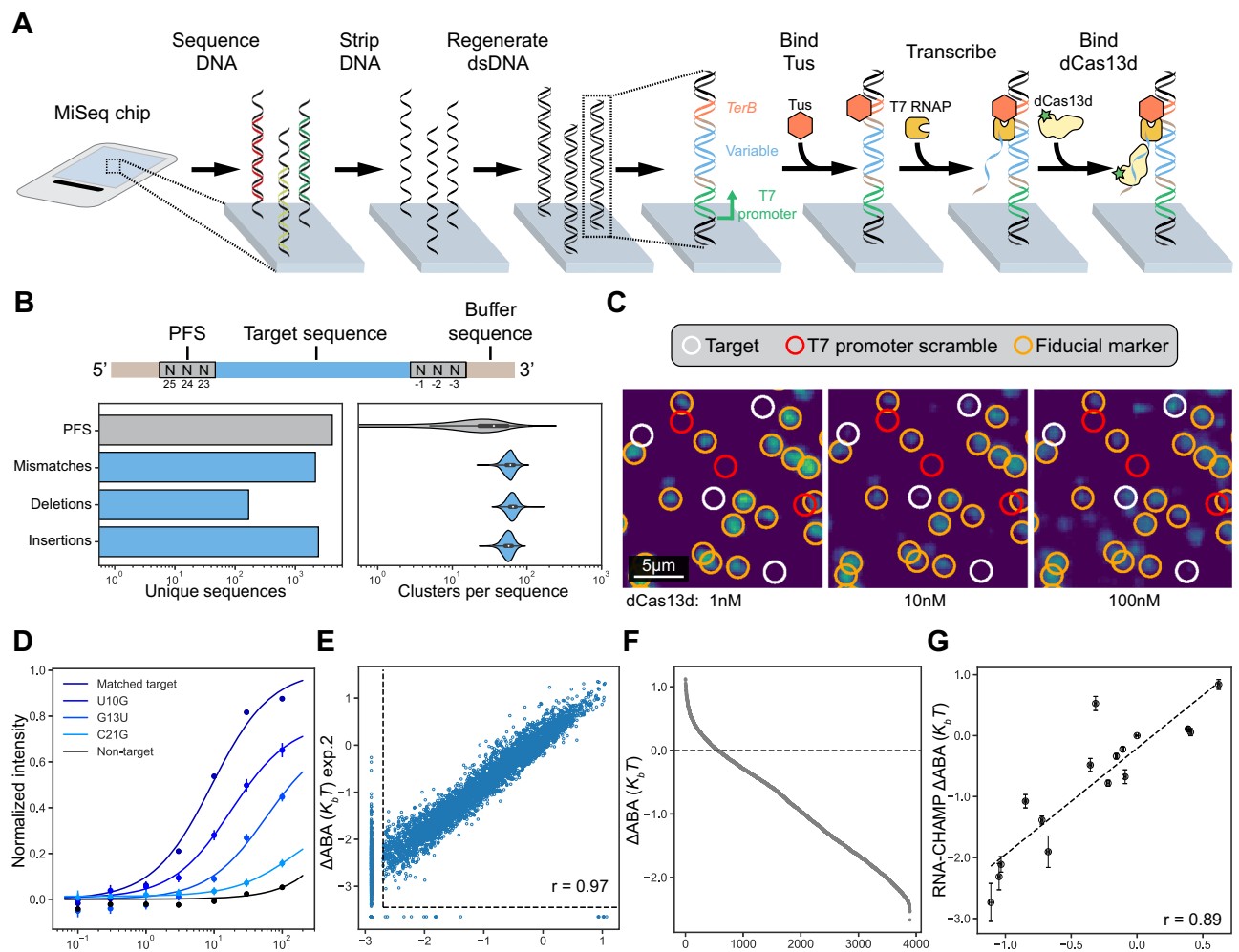

**Fig. 1 | Massively parallel protein-RNA profiling via RNA-CHAMP. A** RNA-CHAMP workflow. DNA is regenerated on the surface of a sequenced MiSeq chip and is transcribed with T7 RNA polymerase (RNAP). Tus retains T7 RNAP and the associated transcript on the DNA. Fluorescent dCas13d is incubated in the chip and the chip surface is imaged. The variable DNA region (blue) is flanked by a fixed sequence (light brown) to maintain the same context. The Tus binding site is labeled in orange. **B** Top: schematic of the RNA library. The 22-nucleotide target RNA (blue) is flanked on both ends by three random nucleotides (PFS, gray) and buffer sequences (light brown). Bottom: summary of the unique DNA sequences in the synthetic library (left), and the number of clusters observed via NGS for each unique library member (right). The violin plot illustrates one of the replicates for target #1, with sample sizes of $n = 2083$ for PFS, $n = 2144$ for mismatches, $n = 167$ for deletions, and $n = 2414$ for insertions. The box extends from the first quartile (Q1) to the third quartile (Q3) of the dataset, featuring a median line. The whiskers are defined as Q1-1.5IQR and Q3 + 1.5IQR, where IQR is the interquartile range of the data. **C** Fluorescent images of the chip surface after incubating with increasing Cas13d concentrations. White circles: library clusters. Red circles: scrambled

promoters that cannot produce RNA. Orange circles: fiducial markers used for image alignment. RNA-CHAMP experiments were conducted in duplicate. **D** Quantification of fluorescent intensities for the indicated mismatch sequences. For example, U10G indicates a U to G substitution at the tenth position in the target RNA. Solid lines are fit to the Hill equation without cooperativity. Data are presented as median ± S.D. from all cluster intensities, with sample sizes of $n = 26,760$ for matched target, $n = 86$ for U10G, $n = 88$ for G13U, $n = 85$ for C21G, and $n = 360$ for non-target. **E** Correlation of two independent RNA-CHAMP experiments. Dashed lines denote the limit of detection. Pearson's $r = 0.97$. **F** Rank-ordered graph of the ΔABA for ~4000 library members. The dashed line represents the ΔABA of the matched target (MT). Sequences below our detection limit in (**E**) are omitted. **G** Correlation of the ΔABA and biolayer interferometry (BLI) - determined binding affinities. Error bars are the standard deviation of ΔABA (RNA-CHAMP) from bootstrap analysis, and 95% confidence interval of the fit (BLI) (clusters numbers for RNA-CHAMP can be found in Table S1, three concentrations for BLI). The dashed line is the linear fit of data points. Pearson's $r = 0.89$.

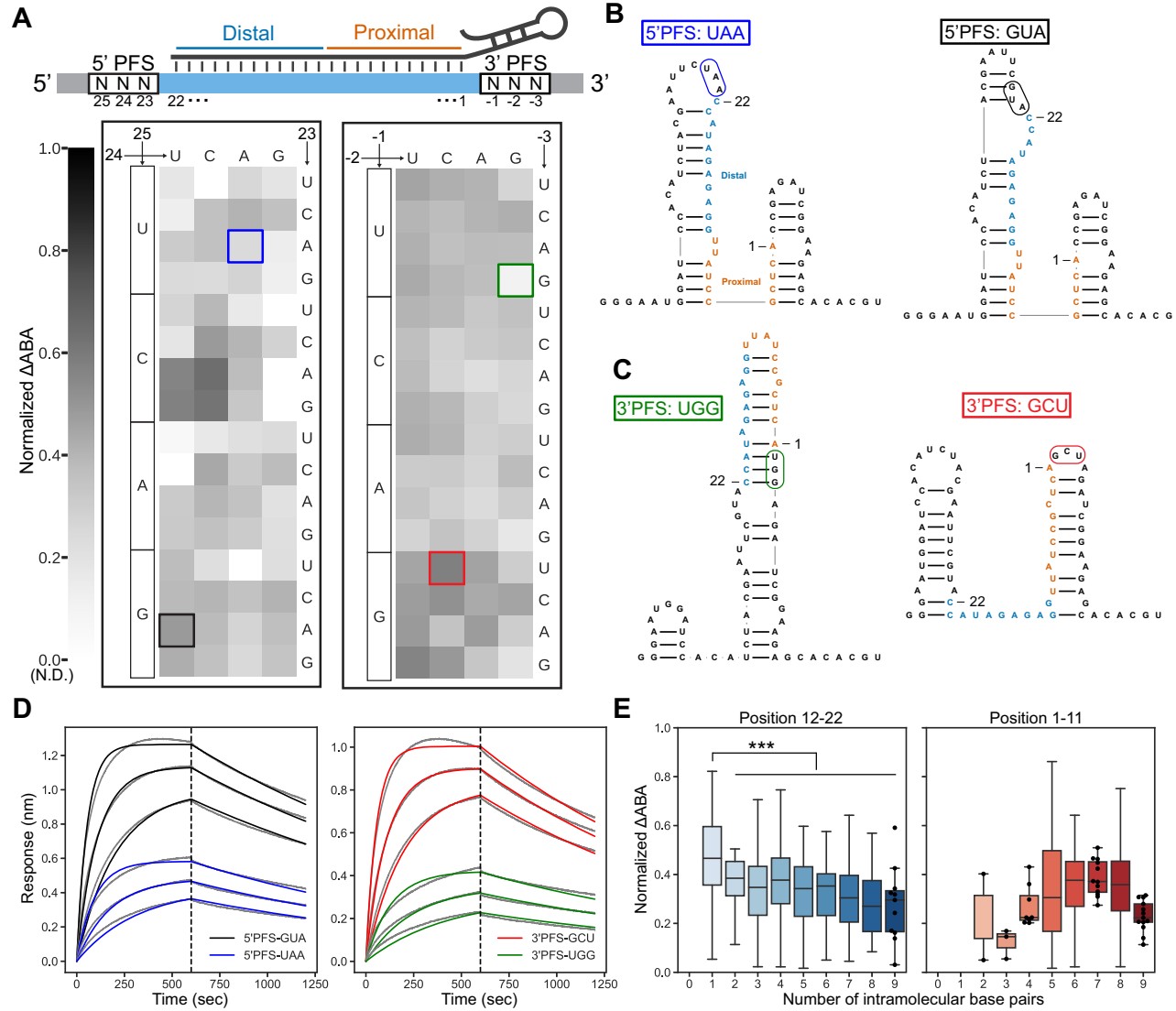

**Fig. 2 | Target RNA structure is a strong determinant of Cas13d binding affinity.**
**A** Top: schematic of the target RNA library. The target RNA is perfectly matched to the crRNA. Bottom: normalized ΔABAs for the 5′ PFS and 3′ PFS. In the plot, each block in the heat map is the mean of all detectable sequences with 5′ PFS (left) and 3′ PFS (right). All sequences were normalized to the scale of zero to 1 for easy comparison between targets. **B** Secondary structure predictions of two illustrative examples for 5′ PFS. Left: a low-affinity target RNA (5′-UAA). Right: a high-affinity target RNA (5′-GUA). The PFSs are boxed in blue and black in (**A**). **C** Predicted secondary structure of a low-affinity PFS (3′-UGG; left), and a high-affinity PFS (3′-GCU; right). The PFSs are boxed in green and red in (**A**). **D** BLI curves of the highlighted PFS sequences in (**B**) and (**C**). Gray lines are experimental curves. Colored lines are the global fit to a 1:1 binding model. **E** Normalized ΔABA of PFS

sequences grouped by their number of intramolecular base pairs within the target region. Graph of the number of intramolecular base pairs in positions 12-22 (Left) and 1-11 (Right). Error bars are the standard deviation of normalized ΔABA. In the distal analysis, sample sizes (n) from left to right were 162, 26, 652, 169, 195, 40, 169, 31, 13. In the proximal analysis, corresponding sample sizes were 2, 3, 9, 103, 160, 16, 1149, 15. Swarm plots were employed when the number of sequences was less than 20. The boxplot extends from the first quartile (Q1) to the third quartile (Q3) of the dataset, featuring a median line. The whiskers are defined as Q1-1.5IQR and Q3 + 1.5IQR, where IQR is the interquartile range of the data. Statistical analysis was conducted using an unpaired two-sided Student's t-test, with significance denoted as ***$p < 0.001$. The corresponding $p$-values for the distal analysis, from left to right, were $P = 0.00012$, 1.2e-23, 2.0e-08, 2.4e-18, 1.8e-07, 1.1e-22, 3.9e-08, 0.00025.

PFS sequence in Fig. 2C has eight proximal and three distal intramolecular base pairs, whereas the 3′-UGG PFS has six proximal and nine distal intramolecular base pairs. We observed that increased intramolecular base pairing in the distal region of the target RNA decreased the ΔABA (Fig. 2E, S3C). In contrast, we did not see any relationship between the number of intramolecular base pairs and the ΔABA in the proximal region (Fig. 2E, S3C). We also compared the base pairing propensity of all suboptimal structures that have a free energy within 1 kcal/mol of the MFE. For example, for an RNA with a predicted MFE of −14.3 kcal/mol, we compare the average number of intramolecular base pairs of all suboptimal RNAs with an MFE of −14.3 to −13.3 kcal/mol. This analysis showed a significant correlation of intramolecular base pairing and binding affinity in the distal region in both targets but

not in the proximal region (Fig S3E, F). Based on these results, we hypothesize that Cas13d prefers to engage the distal end of the target RNA first, and this region must remain partially unstructured for efficient binding (see Discussion). Taken together, we conclude that Cas13d does not have a PFS requirement but prefers to bind target RNAs with unpaired distal nucleotides.

## Cas13d binding is sensitive to mismatches in the distal region of the target RNA
To determine how Cas13d binds off-target RNAs that resemble the target sequence, we constructed a library comprised of 66 single mismatches, 2079 double mismatches, and 2439 insertions & deletions relative to the crRNA within the 22-nt target sequence (target #1)

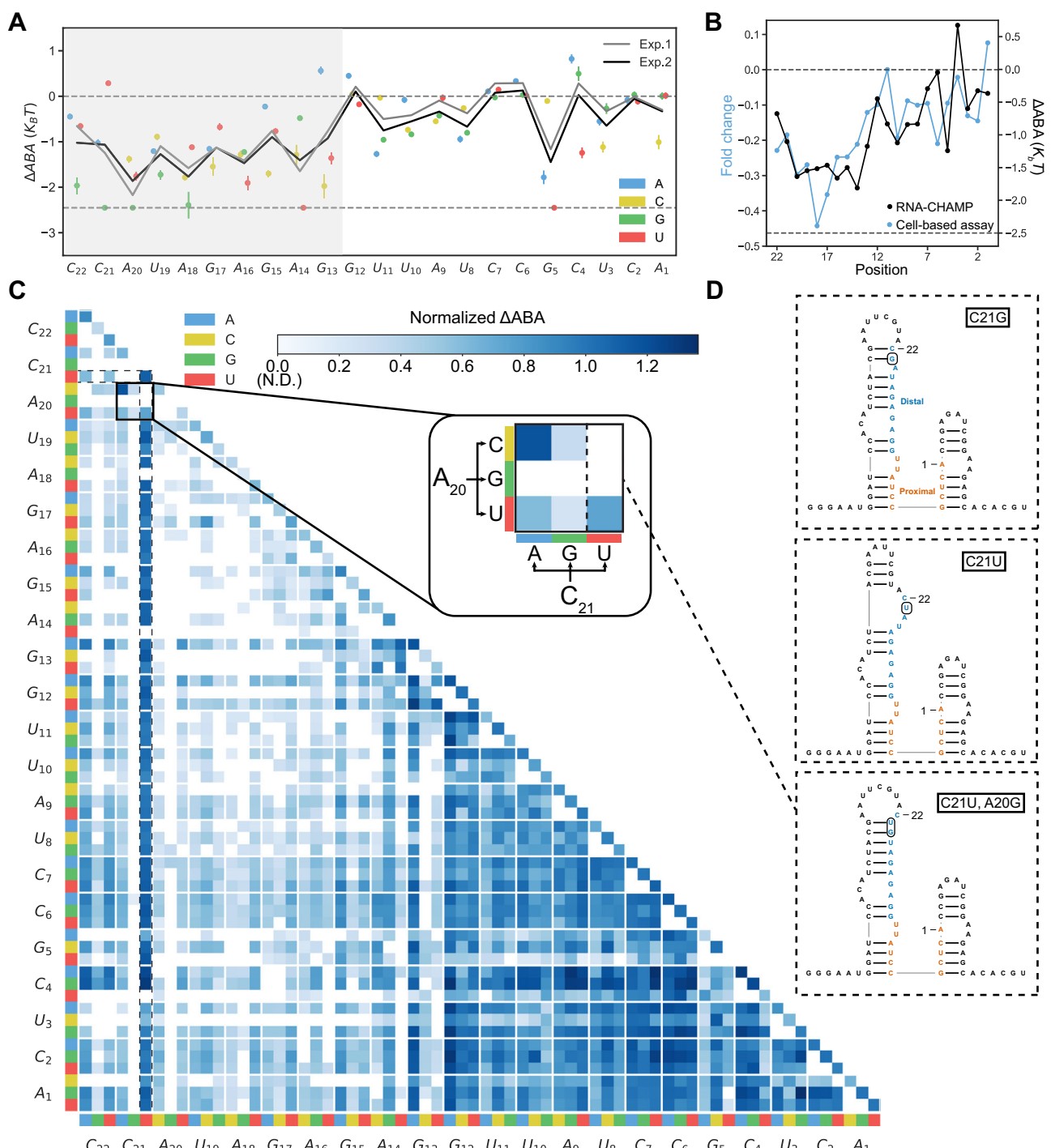

**Fig. 3 | Cas13d binding is sensitive to mismatches in the 5'-region of the target RNA. A** Summary of single mismatch-dependent changes in the ΔABA for two biological replicates. Upper dashed line: matched target ΔABA. Lower dashed line: RNA-CHAMP detection limit. Solid lines: the mean of all three substitutions. Data are the mean ± S.D. from bootstrap analysis. All sequence counts are detailed in the Source Data. **B** Comparison of ΔABA and cleavage of a reporter gene in mammalian cells (data adapted from ref. 32). For RNA-CHAMP, all three possible mismatches were averaged at each position along the target RNA. **C** Normalized ΔABAs of all double mismatched sequences normalized to the matched target. Inset: blowup of all possible mismatches at target positions $A_{20}$ & $C_{21}$. **D** Secondary structure predictions of three illustrative examples. Top: C21G. Middle: C21U. Bottom: C21U, A20G. The mismatches are boxed.

(see Fig. 1B). For all experiments, the 5'- and 3'-PFS remained constant. Of the 4936 library members, we measured ΔABAs for 3893 target RNAs. 1043 sequences didn't significantly change the dCas13d fluorescent signal, even at the highest RNP concentrations. Figure 3A summarizes two biological replicates of the ΔABA for all possible single mismatches. We also measured ΔABAs across a similarly designed library but with different target crRNA sequences (target #3) (Fig. S4). The binding trends were broadly the same across these two libraries.

We first analyzed the impact of a single mismatch between the target and crRNA (Fig. 3A). Mismatches at positions 13-22 significantly decreased the ΔABA. In contrast, single mismatches at positions 1-12 had little to no effect on binding compared to the matched target (Fig. 3A). The identity of the mismatch at the same position led to profoundly different outcomes. For example, a C21U substitution has a similar ΔABA to the matched target, but the binding was virtually undetectable with C21G. The C21U substituted is predicted to match

the structure of the matched target (Fig. 3D, middle). Moreover, C21U creates a G·U wobble base pair with the crRNA, which yields a similar binding affinity to the matched target. C21G, in contrast, creates additional intramolecular base pairs at positions 19-21 (Fig. 3D, top). In a dataset with a different crRNA-target pair, we saw a similar but slightly broader sensitivity region to mismatches at positions 9-20 (Fig. S4A, B). We compared our binding results to a dataset of RfxCas13d RNA cleavage activity reported in mammalian cells (Fig. 3B)[32]. Because this dataset used different target RNA sequences, we compared the mean ΔABA from all three mismatches across two targets to the mean cleavage activity at each position along the RNA target. RNA knockdown efficiency in mammalian cells is reduced when mismatches are in the distal position, analogously to our binding data (Fig. 3B). Overall, Cas13d can tolerate G·U wobble base pairs and shows a strong sensitivity to distal mismatches.

Next, we analyzed the impact of two mismatches on dCas13d binding affinity (Fig. 3C). Binding was largely unaffected if both mismatches occurred in positions 1-12 (dark blue squares in Fig. 3C). We observed multiple instances where the RNA structure drastically changed the ΔABA. Such sequences appear as "stripes" of strong color in Fig. 3C. For example, C21U with an additional substitution (highlighted in dotted line Fig. 3C) does not affect the ΔABA compared to the matched target. However, a second mismatch (A20G) in addition to C21U ablates dCas13d binding due to increased intramolecular base pairing in the distal region of the target RNA (Fig. 3C, D). Overall, we observed that dCas13d prefers unpaired distal RNA sequences. We also observed a strong dependence on RNA structure with the second RNA library (target #3). This target RNA is highly folded, with only bases 20-22 not participating in intramolecular base pairing, reducing overall dCas13d affinity (Fig. S4B, C). For this RNA target, some substitutions (e.g., U2A, G4U) relax the proximal to center region of the target RNA (positions 1-13) structure and result in an increased binding affinity relative to the matched target (Fig. S4C). Cas13d binary structures suggest that positions 4-8 and 14-20 of the crRNA are solvent-exposed and accessible to the environment[13,14]. We speculate that the center exposed region likely contributes to the increased binding affinity. In sum, local RNA structure dominates Cas13d binding affinity. The distal segment of the target RNA must remain partially unpaired for high-affinity binding.

dCas13d retains a high affinity for targets with proximal insertions or deletions (Fig. S5, S6). However, insertions and deletions at the distal side of the target RNA on both targets were not tolerated (Fig. S5, S6). We also observed a strong effect from RNA secondary structure. For example, inserting a C between positions 19 and 20 reduces the number of intramolecular base pairs at bases 13-20, which increases the ΔABA relative to the matched target (Fig. S5A, B). A G-insertion at the same position leads to undetectably low binding due to newly formed intramolecular base pairings in the distal side of the target RNA (Fig. S5B). We observed similar effects of RNA structure on binding affinity in a second RNA target library (target #3) (Fig. S6A, B). A C-insertion at position 3 exposes the proximal region that retains similar affinity to the matched target, while a U-insertion increases intramolecular base pairing and results in undetectable binding. Taken together, these results again show that Cas13d binding is sensitive to distal alterations and local secondary structure. We speculate that Cas13d has a distal seed region and initiates crRNA-target RNA duplexes starting primarily from the distal region (see Discussion).

### Target RNA base pairing is a quantitative predictor for Cas13d binding affinity

We developed a series of linear regression models of increasing complexity to quantitatively understand how mismatches and RNA structure affect Cas13d binding (Fig. 4A). Unlike machine learning approaches (also considered below), these models can elucidate the mechanism of Cas13d binding to partially matched targets. The simplest model (Model I) assigns a position-specific penalty for each intramolecular base pair in the predicted target RNA structure (see Methods & Fig. S7B)[45]. This model requires a total of 22 adjustable parameters, one for each nucleotide along the target RNA. In Model II, we add the predicted minimum free energy (MFE) of the entire 73-nt transcript RNA to capture the overall secondary structure. Model III encodes sequence changes relative to the matched target using a relative encoding strategy (see Methods & Fig. S7A). Model IV adds the target RNA's MFE as another parameter to the relative encoding. Model V combines the intramolecular base pairing penalty and relative encoding. Finally, Model VI includes all three components: intramolecular base pairing penalty, relative encoding, and the MFE (Fig. 4B, C). We trained each model on half of 4,862 partially matched target sequences across two RNA targets (targets #1 & #3). The resulting model was tested on the withheld half of the sequences in our datasets. After fitting the data, each model's performance was evaluated by Pearson correlation and information loss via Akaike information criterion (AIC) (Fig. 4B, C)[46].

Model I, which only considers intramolecular target RNA base pairing within the 22 nt target sequence results in a Pearson's $r = 0.51$. Adding the MFE—a measure of the overall structural stability—only weakly improved the correlation and AIC, indicating that local RNA structure is more important than its global stability. Relative encoding has a lower AIC and a Pearson's $r = 0.65$, performing better than the structure-only model. Finally, combining structural features with relative encoding (Models V) improves both the AIC and Pearson's $r$ to 0.75. Adding the MFE (model VI) slightly improved the AIC, indicating that position-specific mismatches and intramolecular base pairing propensity are sufficient to describe most of the variance in the ABAs. (Fig. 4B–D). We also trained a convolutional neural network machine learning (ML) model on the data (Fig. S7C). Despite having a much larger number of adjustable parameters, the ML model is only marginally better than model VI (Pearson's $r = 0.77$). Since the ML model's parameters are not easily interpretable, it doesn't reveal the mechanisms of RNA binding. Therefore, we dissect Cas13d binding affinities using Model VI below.

We first compared the average penalty for mismatches and indels along the 22 nt target sequence (Fig. 4E, top). Cas13d binding is heavily penalized with mismatches or indels at positions 13-22 along the target RNA. In contrast, mismatches at positions 1-12 only minimally decreased the ΔABA. Likewise, intramolecular base pairing within the target RNA nucleotides 14-22 reduces the ΔABA and is heavily penalized by the model (Fig. 4E, bottom). Intramolecular base pairing within positions 1-13 slightly reduced the ΔABA in the model. Based on these results, we conclude that distal positions 12-22 of the crRNA-target RNA duplex act as an internal "seed" where Cas13d initiates target RNA recognition (see **Discussion**).

### Proximal mismatches suppress Cas13d's nuclease activity

Next, we tested how mismatches affect Cas13d's cleavage activity. We measured the cleavage rates of nineteen single mismatched target RNAs that have also been assayed via RNA-CHAMP (Table S2). Time-dependent cleavage of a reporter RNA (5'-6-FAM-UUUUU-Iowa Black FQ-3') can be followed via an increase in the FAM signal after the fluorophore is released from the quencher (Fig. 5A)[28]. The cleavage rate is monitored via the initial slope of the time-dependent fluorescent signal. Cleavage rates were generally correlated with ΔABAs, with two distinct populations (Fig. 5B–D). Proximal mismatches (i.e., C2G, C4A, and C7A) did not impact RNA binding but only weakly cleaved the reporter RNA. In contrast, distal mismatches decrease both the binding and cleavage rates (Fig. 5B–D). We hypothesize that mismatches at proximal positions disrupt the protein-RNA interface required for activation of the HEPN domain.

Next, we assayed key aspects of our mechanistic insights with *Ruminococcus flavefaciens* Cas13d (RfxCas13d), as this enzyme is

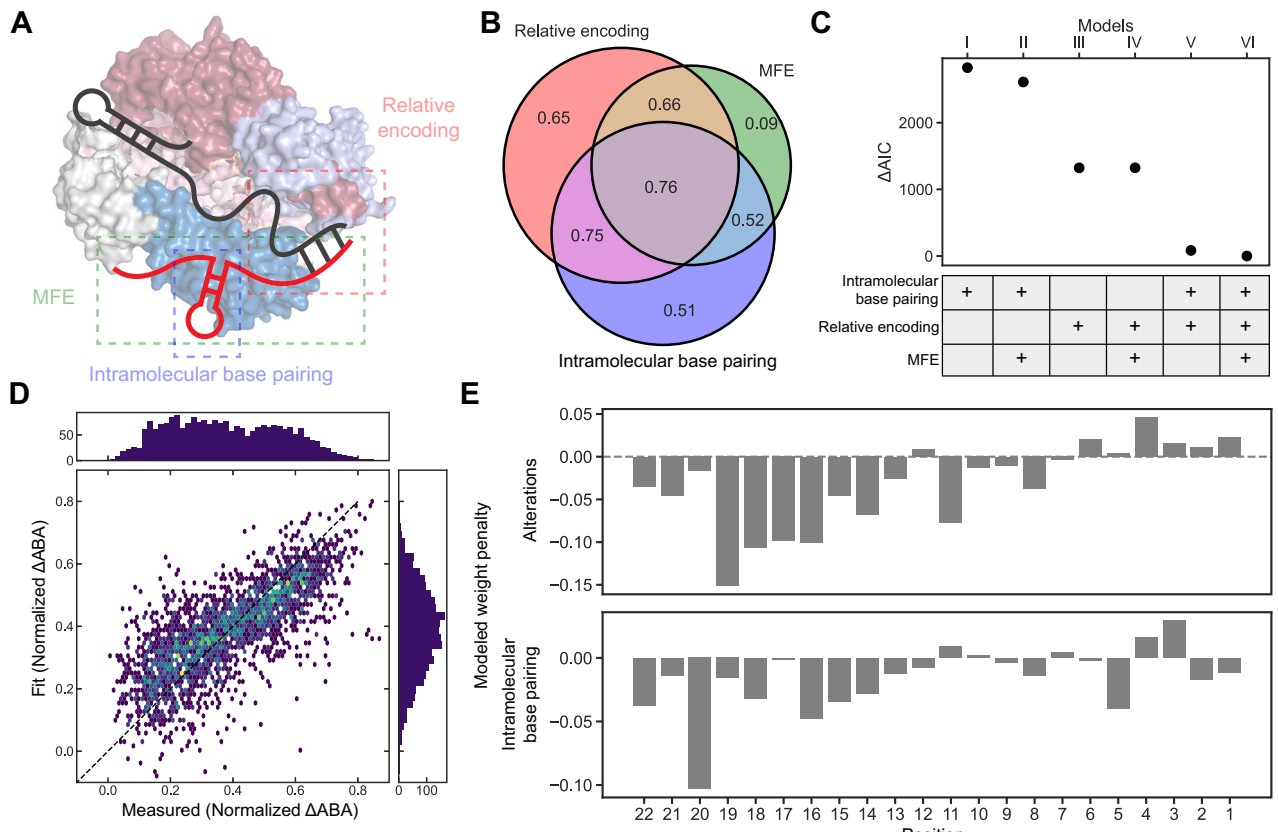

**Fig. 4 | Modeling Cas13d binding. A** Schematic of the three components of our quantitative Cas13d binding models. Relative encoding is the difference between a given sequence and the matched target sequence. The predicted minimum free energy (MFE) of a target RNA is generated by ViennaRNA 2.0[45]. The number of intramolecular base pairing is the count of the RNA base pair in the target region. **B** Venn diagram of Pearson's $r$ correlation coefficients from three main components. Correlation between the measured and predicted data is shown in the Venn diagram. **C** Akaike information criterion (AIC) of the six models used in this study. $\Delta$AIC is the difference between model I-V and model VI. **D** Correlation between the measured and predicted normalized $\Delta$ABAs from model VI. Pearson's $r = 0.76$. **E** The weight penalty of all alterations (mismatches, insertions, and deletions) and intramolecular base pairing by positions in model VI.

widely used for RNA knockdown and engineering applications[5,47,48]. RfxCas13d binding showed a marked sensitivity to RNA structure in the distal end of the target RNA (Fig. S8A), as measured via BLI. This binding sensitivity was strongly correlated between EsCas13d and RfxCas13d (Pearson $r = 0.93$), indicating a similar target recognition mechanism (Fig. S8B). As with EsCas13d, proximal mismatches C2G, C4A, and C7A showed very high binding affinities, but compromised cleavage (Fig. S8C, D). Taken together, we conclude that RfxCas13d and EsCas13d both penalize binding to target RNAs with distal mismatches and structures, and both exhibit a proximal cleavage sensitivity region.

Cas13d's mismatch sensitivity can be exploited to rationally design assays that detect single nucleotide polymorphisms (SNPs) in a target RNA[49]. As a proof of principle of our analysis pipeline, we positioned the SNP in the crRNA-target RNA duplex to differentiate between two SARS-CoV-2 variants of concern (VOC) (Fig. 5E). Here, the matched target is from the spike gene of SARS-CoV-2. The G → A single nucleotide polymorphism (SNP) differentiates the original "Wuhan" strain and the Delta VOC. We designed two crRNAs: the first places this SNP within the binding sensitivity region (crRNA-1; position 17), and the second is in the cleavage sensitivity region (crRNA-2; position 1). Both crRNAs reduce Cas13d cleavage ~5-fold for the D950N RNA (Fig. 5F, G & Table S2). Next, we measured the binding affinity of Cas13d with crRNA-1 and crRNA-2 to both the Wuhan and Delta variants using BLI. As expected, crRNA-1 RNPs only had a weak affinity for the D950N RNA (Fig. S8E, F). In contrast, crRNA-2 RNPs had a comparable binding affinity for both target RNAs (Fig. S8E, F). To confirm that the less efficient cleavage of the Delta variant is not due to RNA

structural changes, we analyzed the predicted MFE structure. The SNP in this sequence doesn't alter the RNA structure (Fig. S8G). As expected, the cleavage rate of the crRNA that matches the Delta sequence is statistically indistinguishable from the cleavage rate of the original matched target crRNA (Fig. S8H, I). The results confirm that the SNP indeed alters the cleavage activity, and the effect is due to the position of the mismatch relative to the crRNA. These results demonstrate that our analysis pipeline can be used to design Cas13d-based diagnostics that distinguish between SNPs by precisely positioning the expected mismatched positions relative to the crRNA.

## Discussion

RNA-CHAMP is a massively parallel platform for probing protein-RNA interactions on used NGS chips. Unlike earlier approaches, CHAMP does not modify any Illumina hardware and is compatible with modern sequencers and chip configurations[35–38]. Imaging biomolecules on upcycled NGS chips can be adapted by any laboratory with a commercial fluorescence microscope that is capable of either TIR- or epi-illumination and a wide-field camera[41,44,50]. In addition to profiling protein-DNA and protein-RNA interactions, related methods have been adapted for peptide display and other imaging applications[51–53]. We envision that the high optical quality and surface passivation of commercial Illumina chips will extend to massively parallel single-molecule imaging.

Using RNA-CHAMP and quantitative modeling, we show that Cas13d has a "seed region" that prefers a relaxed structure at the distal end of the target RNA (Fig. 6). This region is analogous—but not functionally identical—to the PAM-adjacent seed found in Cas9 and

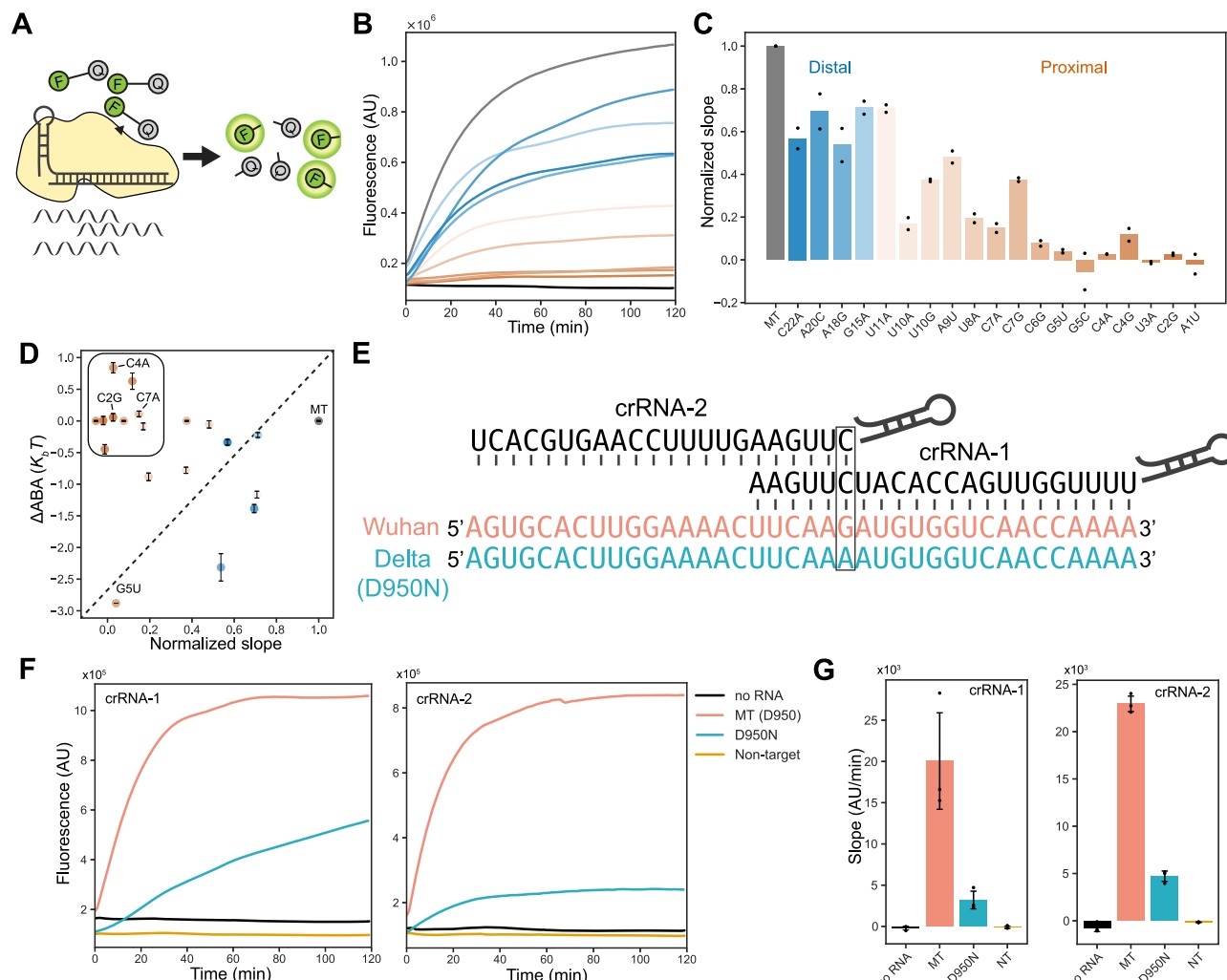

**Fig. 5 | Proximal mismatches limit Cas13d cleavage activity. A** Schematic of the collateral cleavage assay. **B** Fluorescent cleavage time courses for matched target and nine representative mismatched target RNAs. Blue lines are distal mismatched sequences (positions 12-22). Orange lines are proximal mismatched sequences (positions 1-11). **C** The initial slope of 19 mismatched sequences. Slopes are calculated by the fluorescence changes during the first 20 minutes of the cleavage reaction and normalized to the matched target. Data are shown as mean and S.D. from two replicates. **D** Correlation of the cleavage slope with binding affinity

($\Delta$ABA). A subset of target RNAs retain strong binding but are cleavage-inactive (boxed region). Data are shown in mean ± S.D. from bootstrap analysis (y-axis). All sequence counts are detailed in the Source Data. **E** Schematic of mismatch-defined differentiation between SARS-CoV-2 variants of concern (VOC). **F** Fluorescent cleavage time courses for SARS-CoV-2 Wuhan and Delta VOCs. **G** The initial slope of the trace in (**F**). Slopes are calculated by the fluorescence changes during the first 20 minutes of the cleavage reaction. Data are shown as mean and S.D. from three replicates.

DNA-binding CRISPR enzymes[54–58]. The impact of the Cas13d seed is especially profound when the target RNA is perfectly matched with the crRNA. Strong intramolecular base pairing due to changes in the PFS reduces Cas13d binding by over 3-fold relative to a perfectly matched target. Our results highlight that future studies must also consider how the target RNA structure changes enzyme activity. Mismatches can increase the binding affinity when they coincidentally relax intramolecular base pairing within the target. By separating the effect of RNA structure on binding and cleavage, our results explain prior observations that minimal secondary structure in the target RNA correlates with higher cleavage activity in bacterial and mammalian cells[4,30,32,34].

Here, we show that Cas13d binding to the target RNA is penalized when the distal region is structured or is mismatched relative to the crRNA. Target RNAs with the distal region occluded by intramolecular base pairing show significant binding defects. Structures of the binary RspCas13d- and EsCas13d-crRNA complexes reveal a solvent-exposed spacer region in positions 4-8 and positions 14-20 relative to the crRNA[13,14]. Based on the large effects of intramolecular base pairing in the distal position (positions 14-20), we hypothesize that Cas13d

initiates target recognition in this distal region (Fig. 6). Structure of ternary EsCas13d complex suggests that the helical-1 domain has the largest conformational shift compared to other subdomains. Helical-1 domain residues K376, N377, G379, K443, and Y447 are centered around the proximal region of the crRNA (positions 3-6). Mutating residues K376, K443, and Y447 to alanine fails to activate the HEPN domains[13]. Disruption of the protein-RNA interface by either a mismatched base pair or helical-1 amino acid mutations inactivates the nuclease. This indicates that the interaction of the proximal crRNA region and the helical-1 domain is critical for nuclease activation. Further kinetically resolved structural studies will be required to elucidate the mechanisms of target recognition, RNA duplex propagation, and HEPN nuclease activation.

We separately dissect RNA binding and cleavage to reveal that a subset of mismatched sequences can bind with high affinity but fail to activate the nuclease domain (Fig. 6). Cas13d requires base pairing in positions 1-6 to activate its nuclease activity. We leverage this sensitivity to develop guides that can discriminate between circulating SARS-CoV-2 variants. Similarly, LbuCas13a positions 5-8 are critical for

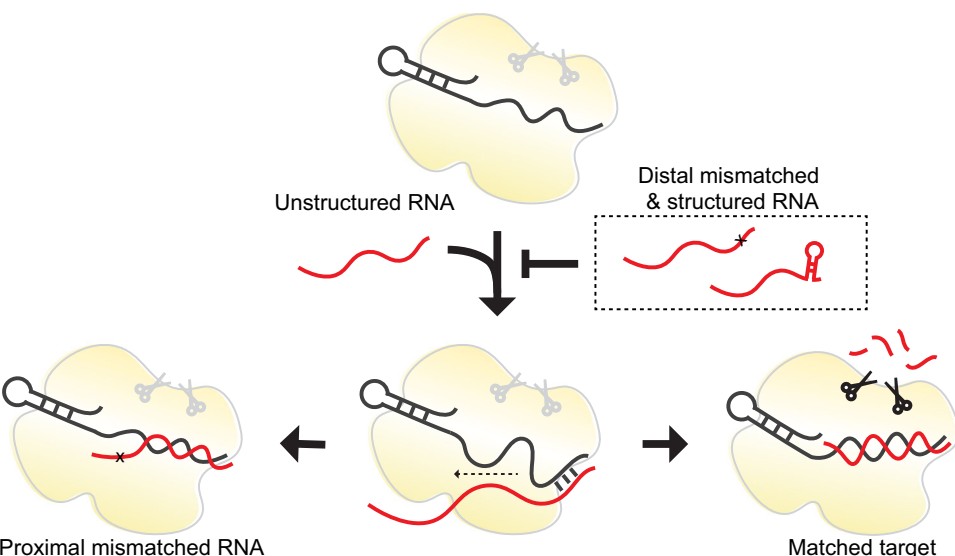

**Fig. 6 | Cas13d binding and nuclease activation follow distinct rules.** Cas13d binding is penalized by distal RNA structures and mismatches. After initial distal recognition, the RNA duplex forms from the distal positions to the proximal positions. A mismatch in the proximal region fails to activate the nuclease activity, leading to a catalytically inactive enzyme. Matched target sequences that form a complete RNA duplex activate the nuclease activity.

cleavage but not binding[31]. This may act as an additional mechanism to suppress nuclease activation and subsequent cell death in prokaryotic hosts. Mismatch-dependent cleavage inactivation may be a universal feature of type VI effectors.

We conclude that Cas13d binding and nuclease activation are governed by distinct spacer-target regions. Mismatches and structural elements in the distal region inhibit binding, whereas proximal mismatches block nuclease activation. These effects, along with the biophysical models developed here, can be selectively used to fine-tune knock-down efficiency in cells by programming mismatches along the crRNA-target RNA duplex. A similar approach has been used to fine-tune CRISPRi with nuclease-dead Cas9 in mammalian cells[59]. In addition, a complete understanding of Cas13d binding and activation can be used for sensitive SNP detection in CRISPR diagnostics (Fig. 5)[15]. More broadly, quantitative studies of RNA-binding CRISPR enzymes must consider the impact of RNA structure on target binding and nucleolytic activity. The structural basis for type VI nuclease activation and the implications for gene editing and prokaryotic immunity are exciting areas for future research.

## Methods
### Oligonucleotides and DNA libraries
Primers, protospacer flanking sequence DNA libraries, crRNAs, and target RNAs were purchased from IDT. Mispaired target DNA oligonucleotide libraries were purchased from Twist or GenScript. DNA libraries for probing the protospacer flanking sequence (PFS) were generated via PCR amplification (Q5 High-Fidelity 2X Master Mix, NEB) of a 6N-oligo ordered from IDT with primers JK044 and JK045 (Table S3). These mixed based oligos included three randomized bases on either end of the target RNA. After PCR, Illumina adapters and sequencing primer attachment sites were added for downstream next-generation DNA sequencing (NGS). Final amplified libraries were constructed as 5'-P5-SP1-buffer sequence-T7 promoter-PFS-target-PFS-SP2-TerB-P7-3'. P5 and P7 are Illumina adaptors, while SP1 and SP2 are Illumina sequencing primers. For mispaired DNA oligonucleotide libraries, we designed a custom oligonucleotide DNA pool (purchased from Twist or GenScript). The DNA pool was PCR amplified using primers JK044 and JK045. These primers also added adapters for Illumina-based sequencing. The PFS and mispaired DNA libraries were pooled and sequenced on a conventional MiSeq instrument using a 150-cycle reagent kit v3 (Illumina). To prevent data loss due to sequencing a low diversity library, we also spiked in sheared human cDNA and PhiX DNA to a total of 50% of the sequencing run.

### Protein Expression and Purification
*Eubacterium siraeum* Cas13d (EsCas13d) was subcloned into a pET19-based plasmid with an N-terminal 6xHis-TwinStrep-SUMO fusion to generate plasmid the pIF1023 from pET28a-MH6-EsCas13d (Addgene #108303). The nuclease-dead variant (dCas13d) was generated by introducing the following mutations into the HEPN active site: R295A/H300A/R849A/H854A. The SNAP-tag was added at the N-terminus of dCas13d (pIF1024). *Ruminococcus flavefaciens* Cas13d (RfxCas13d, or CasRx) was subcloned into a pET19-based plasmid with an N-terminal 6xHis-TwinStrep-SUMO fusion to generate the plasmid pIF1034 from pET28b-RfxCas13d-His (Addgene #141322). The nuclease-dead variant (dRfxCas13d) was generated by cloning the R295A/H300A/R858A/H863A mutations into the HEPN active site.

Catalytically active and nuclease-dead variants of EsCas13d and RfxCas13d were purified using the same protocol. Briefly, the over-expression plasmid was transformed into BL21 star (DE3) cells (Thermo Fisher). Cells were inoculated in LB containing carbenicillin to OD600 - 0.7 and induced with 200 mM isopropyl β-d-1-thiogalactopyranoside (IPTG) at 18 °C for 18 h. Cells were then pelleted, resuspended in lysis buffer (50 mM HEPES pH 7.4, 500 mM NaCl, 1 mM EDTA, 5% glycerol, 0.1% Tween-20, 1 mM DTT, cOmplete-EDTA-free protease inhibitor cocktail (Sigma Aldrich), 1 mg ml⁻¹ lysozyme, 2.5 U ml⁻¹ DNaseI, 2.5 U ml⁻¹ salt active nuclease), and lysed completely by sonication. Clarified lysate was applied to a Strep-Tactin Superflow gravity column (IBA Life Sciences). The Strep-Tactin resin was washed with 20 column volumes (CVs) of wash buffer (50 mM HEPES pH 7.4, 500 mM NaCl, 5% glycerol, 1 mM DTT), eluted with 5 CVs of elution buffer (50 mM HEPES pH 7.4, 500 mM NaCl, 10% glycerol, 5 mM D-desthiobiotin, 1 mM DTT), and then concentrated by a spin concentrator (Amicon 30 kDa cutoff Ultra 15, Millipore). Concentrated samples were incubated with homemade SUMO protease for tag cleavage, and with SNAP-surface 488 dye (NEB), if used for RNA-CHAMP experiments at 4 °C for 20 h. Samples were then further purified by a size-exclusion column (Superdex 200 Increase 10/300 GL, GE Life Sciences) using SEC buffer (50 mM Tris-HCl pH 7.5, 500 mM NaCl, 10% glycerol, 2 mM DTT).

For ribonuclear protein (RNP) reconstitution, purified dCas13d was incubated with a six-fold excess of CRISPR RNA (crRNA; from IDT)

at 37 °C for 1 h in RNP buffer (50 mM Tris-HCl pH 7.5, 100 mM NaCl, 6 mM MgCl₂, 1 mM DTT), and again subjected to size-exclusion column (Superdex 200 Increase 10/300 GL, GE Life Sciences) to further separate the RNP from the crRNA by using RNP SEC buffer (50 mM Tris-HCl pH 7.5, 150 mM NaCl, 1 mM MgCl₂, 10% glycerol, 1 mM DTT). RNP fractions were pooled, spin concentrated (Amicon 30 kDa cutoff Ultra 15, Millipore), flash frozen in liquid nitrogen, and stored at −80 °C.

## RNA-CHAMP

MiSeq chips were collected after sequencing and stored at 4 °C in storage buffer (10 mM Tris-HCL pH 8.0, 1 mM EDTA, 500 mM NaCl) until needed. The chips were placed on a custom-designed microscope stage adapter with integrated microfluidics. The buffer perfusion flow rate was controlled via an automated syringe pump (KD Scientific) and kept constant at 100 μl min⁻¹ for all washing steps. The schematics and CAD files for the microscope stage designs and all additional components are available at https://github.com/finkelsteinlab/RNA-CHAMP.

The chip surface was regenerated after sequencing to remove leftover fluorescent nucleotides and the synthesized strand. The chip was denatured with 500 μl of 0.1 N NaOH and washed with 500 μl TE buffer (10 mM Tris-HCl pH 8.0, 1 mM EDTA). The chip was then incubated with 500 nM regeneration primers IF363 and IF443 (see Table S3) in hybridization buffer (5X SSC [750 mM NaCl, 75 mM sodium citrate pH 7.0], 0.1% Tween-20) for 5 min at 85 °C, cooled to 65 °C over 10 min, cooled to 40 °C over 30 min, and held at 40 °C for 10 min. During the last 10 min at 40 °C, the chip was washed with 1 mL wash buffer (0.3X SSC [45 mM NaCl, 4.5 mM sodium citrate pH 7.0], 0.1% Tween-20) to remove unannealed primers.

For most Cas13d binding experiments, concentration gradients of 0.125 nM, 0.25 nM, 0.5 nM, 1 nM, 2 nM, 4 nM, 8 nM, 16 nM, 32 nM, 64 nM, and 128 nM of Cas13d were sequentially incubated in the chip. At each concentration, Cas13d was incubated for 10 min at 25 °C. Then Cas13d was washed out by 300 μl of protein buffer (40 mM Tris-HCl pH 7.5, 150 mM NaCl, 6 mM MgCl₂, 1 mM DTT, 0.1% Tween-20, 0.2 mg/ml BSA). The fluorescent images were acquired on a TIRF microscope as previously reported[41]. After every imaging experiment, chips were treated with protease K (80 units/ml) diluted in TE buffer (10 mM Tris-HCl, 500 mM EDTA) for 30 min at 42 °C.

## RNA-CHAMP data analysis

Raw images are run through a CHAMP alignment and intensity calculation pipeline[41]. To background subtract the fluorescent buildup on the surface of the chip, the signal from clusters without a T7 promoter was subtracted from the signal for clusters corresponding to RNA library members. Sequences that were represented by five or more physical RNA clusters were globally fit via the Hill equation without cooperativity to calculate the apparent $K_d$, $I_{max}$, and $I_{min}$:

$$I_{obs} = \frac{I_{max} - I_{min}}{1 + \frac{K_d}{x}} + I_{min} \tag{1}$$

Where $I_{min}$ is the minimum intensity for the fit, $I_{max}$ is the maximum intensity for the fit. $x$ is the concentration, and $I_{obs}$ is the observed intensity. To prevent over-interpretting the fitting result, sequences that showed maximum fluorescence intensities below 20% of the matched target intensity were considered "weak binders" and not included in our analysis. Only sequences within our detection limit were included in the analysis (e.g., data within the dashed line in Fig. 1E). We report these as not determined (N.D.). Apparent $K_d$ was transformed to change in the apparent binding affinity (ΔABA) by $\log\left(\frac{K_{d(mt)}}{K_{d(s)}}\right)$, where $K_{d(s)}$ is the Apparent $K_d$ of a library sequence and $K_{d(mt)}$ is the apparent $K_d$ of the matched target. Finally, all experiments were repeated two or more times. For RNA structure prediction, RNA structures were predicted by RNAfold from ViennaRNA[45] using default settings.

## Biolayer Interferometry

Binding kinetics were assessed via biolayer interferometry on an Octet RED96e (FortéBio). Biotinylated RNAs in Table S4 was immobilized on streptavidin biosensors (FortéBio). The biosensors were subsequently submerged in protein buffer (40 mM Tris-HCl pH 7.5, 150 mM NaCl, 6 mM MgCl₂, 1 mM DTT, 0.1% Tween-20, 0.2 mg/ml BSA) containing dEsCas13d RNP complex at concentrations of 100 nM, 50 nM, and 25 nM for 600 seconds to measure association. The biosensors were transferred to protein buffer for 600 seconds to measure dissociation. The dRfxCas13d experiments were conducted using a single concentration (25 nM). We also acquired the signal from a reference sensor without any dCas13d RNP. This trace was treated as a baseline and subtracted from all other association and dissociation curves. The $k_a$, $k_d$, and $K_d$ values were calculated from global fitting to all the binding curves by using Octet data analysis software v11.1. All BLI measurements are summarized in Table S1 and the Source Data.

## Collateral cleavage fluorescent assay

Catalytic active Cas13d was purified as described above. 50 nM of Cas13d were incubated with 50 nM of poly-U reporter (5′−6-FAM-UUUUU-Iowa Black® FQ-3′, IDT) and 5 nM of the indicated target RNA (IDT; Table S4). The reaction was incubated in a 96-well plate in the RT-PCR system (ViiA 7) at 25 °C. Fluorescent intensities were detected every minute for a total duration of 120 min. Technical duplicates were done in every plate, and two or three biological replicates were done for each sequence. The mean of the technical duplicate of one experiment was shown in the plot. A subset of the mismatch sequences was produced by in vitro transcription (IVT). The IVT templates were generated by hybridizing two oligos that contains the T7 promoter sequence (IDT; Table S3). IVT reactions were performed by using the HiScribe T7 High Yield RNA Synthesis kit (NEB). IVT products were subsequently purified by RNeasy mini kit (Qiagen). The initial slope at the 20 min time point was calculated to quantitatively compare the cleavage activity. All fluorescent cleavage data are summarized in Table S2 and the Source Data.

## Computational modeling

To extract mechanistic insights into off-target RNA binding, we created generalized models across all target RNA experiments. First, all ΔABAs were normalized to be between 1 and 0 for the upper and lower detection limits, respectively. Model I solely considers intramolecular base pairing across the 22 nucleotides target RNA according to the function below. The RNA structure was predicted by ViennaRNA[45]. The model adjusts the 22 parameters $a_i$, one for each base in the target RNA sequence:

$$f_{BP}(i) = \begin{cases} 1, & \textit{if position i was based paired with other bases} \\ 0, & \textit{Otherwise} \end{cases}$$

$$\text{Model I}: \widehat{k_{\Delta ABA}} = \sum_{i=1}^{N} a_i * f_{BP}(i) \tag{2}$$

Model II includes an additional term, $g(k)$, the predicted minimal free energy (MFE) in kcal mol⁻¹ of sequence $k$ (predicted by RNAfold from ViennaRNA[45]).

$$\text{Model II}: \widehat{k_{\Delta ABA}} = \sum_{i=1}^{N} a_i * f_{BP}(i) + e * g(k) \tag{3}$$

Model III is the relative encoding-only model and has three main terms that summarize the relative penalties for insertions ($I$), deletions ($D$), or mismatches ($M$) in the target sequence relative to the matched target. As an example, consider a sequence with C2G and U10A alteration compared to the matched target strand, $k_{mt}$. These

operations can be conceptually written as:

$$\{(Mismatch,2,G),(Mismatch,10,A)\}$$

Thus $f_M(2,G)$ and $f_M(10,A)$ would evaluate to 1 and all other inputs for $f_x$ would evaluate to 0. There was a total of 9 parameters for each RNA position: deletion, insertion A, insertion U, insertion G, insertion C, mismatch A, mismatch U, mismatch G, and mismatch C. Here, $i$ denotes the sequence position, $v$ is the altered RNA base identity, and $b, c, d$ are the three sets of adjustable parameters.

$$f_x(i,v) = \begin{cases} 1, & if\ oper\ x\ used\ to\ transform\ from\ k_{mt}\ to\ k \\ 0, & Otherwise \end{cases}$$

$$\text{Model III}: \widehat{k_{\Delta ABA}} = \sum_{i \in I} b_{i,v} {}^* f_I(i,v) + \sum_{i \in D} c_i {}^* f_D(i,0) + \sum_{i \in M} d_{i,v} {}^* f_M(i,v) \quad (4)$$

Model IV includes an additional MFE term, $g(k)$, as previously described.

$$\text{Model IV}: \widehat{k_{\Delta ABA}} = \sum_{i \in I} b_{i,v} {}^* f_I(i,v) + \sum_{i \in D} c_i {}^* f_D(i,0) \\ + \sum_{i \in M} d_{i,v} {}^* f_M(i,v) + e {}^* g(k) \quad (5)$$

Model V is the combination of Model I and Model III that includes both intramolecular base pairing and relative mismatch/indel encoding.

$$\text{Model V}: \widehat{k_{\Delta ABA}} = \sum_{i=1}^{22} a_i {}^* f_{BP}(i) + \sum_{i \in I} b_{i,v} {}^* f_I(i,v) + \sum_{i \in D} c_i {}^* f_D(i,0) \\ + \sum_{i \in M} d_{i,v} {}^* f_M(i,v) \quad (6)$$

Model VI has the following parameters: base pairing, relative encoding, and the MFE of the predicted lowest-energy structure. The normalized ΔABA for partially matched RNAs, $k$, that are related to the matched target, $k_{mt}$, was modeled using a linear combination of the following features. $f_x(i,v)$ denotes different types of sequence alteration and RNA accessibility in the model, where $BP, I, D,$ and $M$ were base pairing, insertions, deletions, and mismatches respectively.

$$\text{Model VI}: \widehat{k_{\Delta ABA}} = \sum_{i=1}^{22} a_i {}^* f_{BP}(i) + \sum_{i \in I} b_{i,v} {}^* f_I(i,v) + \sum_{i \in D} c_i {}^* f_D(i,0) \\ + \sum_{i \in M} d_{i,v} {}^* f_M(i,v) + e {}^* g(k) \quad (7)$$

The weights of the terms $a_i, b_{i,v}, c_i, d_{i,v},$ and $e$ are the adjustable parameters that are used to fit the experimental data and represent the penalties of each operational transformation on altered sequences $k$ in the library.

Ridge regression was used to determine the weights of our parameters to fit the experimental training set. Ridge regression is a variant of linear regression that attempts to minimize the training loss value of the expression:

$$\sum_{k \in T_{tr}} (M(k) - k_{ABA})^2 + \lambda \sum_{\beta \in X} \beta^2$$

Where M is defined to be the model for predicting ABA with all the weights $\beta$ being values in set X = $\{a_i, b_i, c_i, d_i, e\}$. The predicted values from Model M are compared to the measured ΔABA values, $k_{\Delta ABA}$: the smaller the absolute difference between the two the greater the model's accuracy. Ridge regression helps maintain the robustness of linear models and prevents overfitting by penalizing arbitrarily large

weights. The parameter $\lambda$ at which the weight values in the model appear to stabilize is around 1 which was used throughout all models.

Finally, we considered a sequential convoluted neuron network (CNN) model. The model was built on a single Conv2D layer with 64 filters and a kernel size of 54 × 1[60]. Then, a final layer of MaxPooling2D with a pool size of 3 × 1 was added. The CNN model was trained by the same dataset used in the simple linear model. The dataset has 4862 sequences, and only half of the sequences were used for training the model. The model was trained through 1000 epochs and was tested on the rest of the sequences.

## Reporting summary
Further information on research design is available in the Nature Portfolio Reporting Summary linked to this article.

## Data availability
BLI and fluorescent cleavage data is available in the Supplementary Table S1, S2, and Source Data file. RNA-CHAMP data is available in the Source Data file. Source data are provided with this paper.

## Code availability
Source code associated with this work is available on GitHub: https://github.com/finkelsteinlab/RNA-CHAMP.

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

## Acknowledgements

We thank the staff of the University of Texas at Austin Genomic Sequencing and Analysis Facility, Dr. Rick Russell, and members of the Finkelstein laboratory for carefully reading the manuscript. We thank the funding agencies for supporting this work, including the College of Natural Sciences Catalyst award (to I.J.F.), the Welch Foundation (F-1808 to I.J.F.), and the National Institutes of Health (R01GM124141 to I.J.F.). The

content is solely the responsibility of the authors and does not necessarily represent the official views of the National Institutes of Health.

## Author contributions

H.-C.K., J.P., C.-W.C., and I.J.F. designed the research. H.-C.K. performed and analyzed the experiments, wrote the bioinformatics software, and performed biophysical modeling. J.P. implemented the biophysical models. C.-W.C. purified proteins and performed biochemical experiments. I.J.F. secured funding. H.-C.K. and I.J.F. wrote the paper with assistance from all co-authors.

## Competing interests

I.J.F. has filed a patent application titled "Chip hybridized association-mapping platform and methods of use" (US16/622,441), which is currently pending approval. Additionally, H.-C.K., C.-W.C., and I.J.F. have submitted an invention disclosure related to RNA-targeting via Cas13 enzymes. The remaining authors declare no competing interests.
