## [Peer Review File · Nature Communications]

Reviewers' Comments:

Reviewer #1:

Remarks to the Author:

This manuscript by Kuo et al. repurposes the chips used in Illumina sequencing for high throughput biophysical characterization of Cas13d binding on many RNA targets with known sequences. These sequencing chips contain clusters of DNA molecules of varying DNA sequences, with each cluster containing many DNA molecules of a particular sequence. These clusters are then subjected to in situ transcription which is then purposefully stalled by a Tus protein (a known RNA polymerase staller) bound to the flanking *terB* sequence. This stall not only stalls the transcription but causes the transcribed RNA to be tethered to the stalled complex. Thus, each DNA in the cluster has a dangling transcribed RNA whose transcription was deliberately stalled by the Tus. Therefore, you get a stalled RNA molecule from each DNA molecule in the cluster, thereby giving multiple RNA molecules (of known sequence) for each cluster. Fluorescently tagged RNA binding proteins (like Cas13d) can then be added (& later removed) to this chip containing clusters of different RNA molecules. With the addition and removal of this fluorescently tagged protein, the changes in fluorescent signal over these clusters report on the biophysics of interactions between the Cas13d and RNA (of different known sequences). Authors call this pipeline CHAMP. These kinds of pipelines are not new and have been reported earlier including for CRISPR systems. The pioneers of these pipelines include Finkelstein group itself (authors of this manuscript) and the Greenleaf group, amongst others. These types of pipelines are a powerful method for high throughput profiling of the impact of nucleic acid sequences on the protein-nucleic acid interactions. I think they are even more helpful for the RNA because these sequences guide RNA's secondary structures. Therefore, you can profile not only the sequence dependence but also secondary structure dependence on the protein-nucleic acid interactions.

The experiments reported in the manuscript are expertly performed and reasonably clearly communicated. The number of experiments and different types of guide RNA and combinations in its sequence tested, ranging from those in protospacer flanking sequence (PFS) to those in PFS-distal and PFS-proximal region, are pretty high underlying the strength of the observations. The authors then combined this extensive experimental data with modeling that would better help understand the underlying structure of the comprehensive experimental data. So, I commend the authors for their hard work and thorough investigation. I am quite positive about this manuscript's acceptance for publication in Nature Communications. But there are some comments (one major and a lot of minor ones) whose addressal will improve the interpretability and communication of their results. These comments are:

1. [Major comment]: While authors report controls to show that results from CHAMP can be recapitulated on other biophysical assays like the BLI. But one of my concerns is if all RNA molecules in a cluster are transcribed to the same length. What if the RNAP unloads before it is stalled by Tus. I am guessing such premature unloading of RNAP will cause such RNA to detach from the DNA, allowing for a new round of transcription to occur. Therefore, only RNA molecules whose transcription reaches till Tus will remain tethered to the DNA molecule. Perhaps authors can explain this part in their methods section. The second concern is the varying degrees of transcription efficiency for different sequences. All authors' analyses assume that roughly the same number of RNA molecules will be in each cluster. But I am guessing transcription would have a sequence dependence, thereby likely giving different numbers of RNA molecules per cluster, which would then affect the strength of the fluorescent signal for each cluster. Do authors do additional controls (other than BLI controls) to account for this effect?
2. While the manuscript is clearly written, I think it is a bit too heavy on Biophysics and can be simplified further to increase its reach to broader audience. A non-biophysical audience may find it difficult to understand the whole manuscript fully. For this, I suggest the following
 - A single sentence in the main text explaining the range of values ΔABA takes and what these values mean for Cas13d and RNA binding.
 - Figure 1B and 1G can be moved to SI figures to increase the clarity of main text figures. These two panels are not needed for main text figures.

- The color scheme should be improved to avoid confusion. E.g., Cyan is used for term sequence as well as target sequence in Figure 1B. Orange color is used for PFS in Figure 1B and for fiducial circles as well. These minor color inconsistencies reduce the ease of perusing the figures, so I suggest authors develop a better color scheme that would be consistent and non-confusing throughout the manuscript.
- Title of Cas13d requires a partially unstructured target RNA should be better titled as Cas13d has no strong PFS require but requires a partially unstructured target RNA in distal region.
- “This preference, however, only partially explains the wide range of Δ ABAs for these datasets”. I did not fully understand the logic of this statement in main text. Can you please explain why this only partially explains? It is clearer from the subsequent analysis, but as a standalone statement, I did not understand how the sequence only partially explains the wide range of Δ ABA
- “no statically significant” should be no statistically significant
- Title of Cas13d is sensitive to mismatches in the distal region of the target RNA is better titled as Cas13d binding is sensitive to mismatches in the distal region of the target RNA, especially ones that form secondary structures there.

3. Lack of sufficient discussion on the following significant aspects of Cas13d:

- How do the preferences observed for Cas13d’s binding and cleavage relate to its structure?
- Most CRISPR enzymes like Cas9 strongly depend on PAM-proximal region for binding and the PAM-distal region for cleavage, where this dependence is reversed for Cas13d; can you speculate about this reversal? Is it purely based on its structure? Or is there an underlying biological significance for this unusual (from other CRISPR enzymes) dependence?
- Maybe authors can more strongly emphasize that these kinds of high throughput profiling for sequence dependence are even more critical for RNA because they help us understand the importance of both nucleotide identities and secondary structures they create.
- CRISPR enzymes’ collateral nuclease activities have been repurposed to amplify the detection of specific nucleic acid sequences, thereby helping create virus diagnostic tools. But I see that these tests have not taken off. Most of the tests I see are still PCR based. Did these tests not take off? because of {a} Lack of amplification and thus lack of signal {b} Ease of use and applicability. {b} Lack of sensitivity, which authors have shown can be improved by knowledge about the specificity of CRISPR enzymes like Cas13d. Perhaps authors can talk a little more about the future of CRISPR-based nucleic acid diagnostic tests.

Some of these above comments are not critical, and authors may not have space to accommodate them all. Therefore, even though I think these comments will improve the manuscript further, it is okay if the authors cannot accommodate all into the revised manuscript.

Reviewer #2:

Remarks to the Author:

This manuscript introduces a highly innovative method (RNA-CHAMP) to measure the binding affinity of Cas13 enzymes over many RNA targets. Beyond technology development, several important and innovative insights are also discovered using this approach, including (1) Cas13d does not have PFS preferences, (2) Cas13d binding is sensitive to mismatches mainly from distal regions; (3) target RNA base pairing predicts binding affinity; and (4) proximal mismatches do not affect binding but suppresses nuclease activity. These discoveries are uniquely made available with the proposed RNA-CHAMP technology.

Despite these enthusiasms, I have several major concerns:

1, authors use EsCas13d and its dead version (dCas13d), which is different from the widely used RfxCas13d (or CasRx). I will recommend performing RNA-CHAMP experiments on RfxCas13d. The purpose is not to repeat every experiment in RfxCas13d, but rathe to evaluate whether the major conclusions also apply to RfxCas13d, which is used much more frequently than EsCas13d.

2, Figure 2 B,C and Figure 3 D: most of the target sequence (sequence between 5' PFS and 3'PFS) form basepairs. An important finding is that the binding affinity is reversely correlated with the number of mismatches (Fig. 2E). Two associated questions:

- is the RNA structure determined by computational predictions? How reliable is the predictions; i.e., are there multiple possible secondary structural predictions with similar MFE?
- By looking at the examples in Figure 2B and C, it appears the maximum length of consecutive unpaired bases is also critical, as a long consecutive unpaired base appear to have stronger binding (e.g., ACCAU in 5'PFS: GUA in Fig. 2B; and AUAGAGA in 3' PFS GCU in Fig. 2C). Would that feature (maximum length of consecutive unpaired bases) associated with delta ABA value?

3, The main conclusion of Figure 5 A-D is that proximal mismatches do not affect binding affinity but strongly suppresses nuclease activity. Therefore, crRNA2 should have similar binding affinity but weaker cleavage in D905N than Wuhan strain. In contrast, crRNA1, whose mismatch is on the distal region at D905N, should have both weaker binding affinity and cleavage in D905N than Wuhan. Authors only show crRNA1/2 have weaker cleavage in D905N than Wuhan, but no binding comparison is shown, which will be key to confirm the conclusions in Figure 5.

Minor comments:

- If applying the model (Figure 4) to the guides in Figure 5, do the prediction results match the experimental results?

Reviewer #3:

Remarks to the Author:

Kuo et al has advanced their previous high throughput fluorescence method, CHAMP, for studying RNA studies. One notable novelty of the study is the decoupling of binding and cleavage, which distinguishes it from previous investigations. However, to enhance this aspect, the authors should provide further clarification on the rationale behind certain choices and elaborate on the obtained results. For example, it would be advantageous to expand the high-throughput analysis of cleavage beyond the current limited set of nine sequences.

Furthermore, it is important to address certain points to improve the overall understanding of the study. Specifically, the authors should explain the reasons behind a significant number of library members yielding results below the detection threshold. Additionally, clarification is needed regarding the fixed sequences adjacent to the target that introduce secondary structure. Consideration should be given to including a control group without strong secondary structures to serve as a baseline for comparison.

Overall, the manuscript should be rewritten to acknowledge the existing literature properly and avoid giving the impression that certain characteristics of Cas13d are being revealed for the first time. While the study does contribute novel findings, it would be beneficial to strengthen certain aspects by providing clearer explanations, additional information, and expanding the scope of experimentation.

I have attached detailed comments outlining specific areas of concern.

Major comments

1. Concerning the introduction

1.1. The current introduction reads as if RNA-CHAMP is a completely new method. I ask the authors to mention RNA-MaP and their own CHAMP in the introduction (or first results section) instead of in the discussion.

1.2. The introduction does not give a correct picture of what is and what is not already known about EsCas13d. The current text explains that different Cas13-family members have different or no PFS requirements and refers to one study on the effect of mismatches on Cas13d activity. However, some very relevant previous findings specific to Cas13d are missing. For example, the already existing evidence and structural explanation for the lack of a PFS requirement for Cas13d (Yan et al. Mol Cell 2018, Konermann et al. Cell 2018, Zhang et al. Cell 2018) are not mentioned.

Additionally, in previous studies there were already observations of a negative correlation between secondary structure of the target RNA and Cas13d efficiency (e.g. Yan et al. MolCell 2018). This is mentioned later (line 320) but it should be informed in the introduction.

2. Concerning the section "Cas13d requires a partially unstructured target RNA": In line 148, the authors describe that they randomized three nucleotides on both the 3' and 5' end of a perfectly matched target and that they measured the Δ ABA for 1457 different combinations. The other sequences, more than half of the in total 4096 sequences, were below the detection threshold. It would be very informative to know whether these sequences have something in common (e.g. a very strong secondary structure) that explains why they could not be measured or whether there is another explanation for such a large part of the library, with perfectly matched target sequence, being below the detection threshold.

3. Concerning the section "Cas13d is sensitive to mismatches in the distal region of the target RNA"
3.1. In line 190 the authors state that the binding trends were broadly the same across the two libraries. Comparing Fig. 3A and Fig. S4A, I agree that broadly there is a similar trend but there is a difference in where the mismatch-sensitive region starts as also acknowledged by the authors in line 199. Would this be due to the stronger secondary structure and thus weaker binding of target #3 as mentioned later in line 217? What is the expectation for how target-dependent the position of the mismatch-sensitive region is?

3.2. From the results and structural predictions presented in Figure 2 and 3, it becomes clear that the fixed sequences next to the target and PFSs form secondary structures with the target. What is the motivation for placing these sequences within the transcribed RNA and not after the terB and before the T7 promoter site, respectively? The fixed sequence at the 3' side of the target is a sequencing primer but what is the sequence on the 5' side of the target? 3.3. Relatedly, to decouple the secondary structure and sequence effect, a control experiment should be performed where the flanking fixed sequence does not form secondary structures with the target.

3.4. In line 227 the authors state "However, insertions and deletions at the distal side of the target RNA on both targets were not tolerated (Fig. S5, S6)". However, for the target in Fig. S5, many of the insertions decrease Δ ABA only slightly. For the target in Fig. S6 almost all insertions/deletions (also in the proximal region) abolish detectable binding, but as was already mentioned before this target is highly structured and thus very weakly bound by Cas13d. Please comment.

3.5. To strengthen their claim about the effect of secondary structure, could the authors consider providing a plot with Δ ABA versus the number of intramolecular base pairs in the target instead of picking out two examples?

4. Concerning the biophysical model: How does the performance of the presented biophysical model compare to DeepCas13 (Cheng et al. Nat Commun 2023)? DeepCas13 predicts on-target activity while the biophysical model presented by the authors is about binding, but it would be very interesting to see the differences and whether they can be explained by the insights the authors gained by decoupling binding and cleavage activity.

5. Concerning the cleavage data

5.1. Figure 5D, what is the Pearson correlation coefficient? With 9 sequences, of which 3 are deemed outliers (more than half of the proximal mismatch sequences), it is hard to draw a conclusion about general rules for the effect of mismatches on Cas13d cleavage. And, as their conclusion contradicts data from a large-scale Cas13d screen in mammalian cells by Wessels et al. Nat. Biotech. 2020, the authors should compare their cleavage data with the cleavage data (not just binding data) from Wessels et al.

5.2. The authors have chosen only 9 sequences to study cleavage. Please justify why these 9 sequences were chosen. How about the use of NucleaSeq developed in their earlier publication (Jones et al. 2021, Nature Biot)?

6. Concerning the proof-of-principle for SNP detection

6.1. As it is not the first time that Cas13d was used for single nucleotide variation, they should cite Qiao et al. Biotechnol. Bioeng. (2021).

6.2. Both in the introduction (line 75) and in line 291 the authors state that using the insights gained from this study they can rationally design assays to detect single nucleotide polymorphisms (SNPs) in a target RNA. However, it does not become clear what the rationale is behind the design

of the two used crRNAs. In line 296 the authors also mention that the two crRNAs were designed such that the SNP is positioned within a sensitive region of Cas13d activity. Can the authors pinpoint the sensitive region of Cas13d activity? With the claims that distal mismatches strongly decrease binding and cleavage and proximal mismatches inhibit cleavage, Cas13d activity seems sensitive to mismatches regardless of their position within the targetRNA. Why were positions 1 and 17 chosen and would a mismatch at any position work?

6.3. The proof-of-principle nicely demonstrates that two SARS-CoV-2 variants can be distinguished using Cas13d with a crRNA that overlaps with an SNP in the target RNAs. However, an alternative explanation for why this difference occurs is possible. As shown in the RNA-CHAMP assay with the PFS library, the secondary structure of the target RNA plays a large role in determining Cas13d activity. The SNP in the SARS-CoV-2 variants can affect the secondary structure. With the current data, it cannot be ruled out that even a matching guide would cleave the Delta variant less efficiently than the Wuhan variant. Therefore, a control with a matching guide for the Delta variant would help to say with more certainty why there is a difference in cleavage activity between the two variants.

7. In the Discussion, line 325, the authors state "Cas13d requires basepairing in positions 1-6 to activate its nuclease activity." Which data is the basis for this range?

Minor comments

1) In Figure 1A the schematic suggests that the RNA only contains the variable region, while it also contains a fixed sequence of 24-34 nt on both sides of the target. It would be helpful for the reader if these fixed sequences are also indicated in the schematic.

2) In the caption for Figure1B, bottom, left and right are switched.

3) In line 99 the authors state "Importantly, Tus did not bind clusters that lacked terB." However, in Figure S1A, the bar plot does not include a bar for the clusters lacking terB.

4) In line 158 reference 4 is missing.

5) For Figure 3A, please add in the caption what the solid lines represent. Assumably the average of all mutations for each position?

6) For Figure 4E, please add in the caption what the bottom 'Base pairing' penalty is for. Actually, the authors might consider changing the term base pairing (also in the main text) to something like 'internal base pairing' or 'intramolecular base pairing' because the reader might become confused with base pairing between the target and crRNA instead of internal base pairing within the target.

7) In lines 294-296 the authors state: "Here, the matched target is from the spike gene of the original("Wuhan") SARS-CoV-2, but harbors a single G to A substituting (D950N in amino acid sequence) in the position 1 and 17 in the Delta VOC." This sentence is confusing as it suggests that there are two mutations in the Delta VOC with respect to the Wuhan variant, while there is a single mutation which is placed at position 1 and 17 of crRNA-2 and crRNA-1, respectively.

8) The methods section would benefit from a revision as it contains several inconsistencies and textual errors. Among others the following (line numbers refer to the lines in the supplementary file):

8.1. In line96 there is a reference to primers 'JK044' and 'JK045' while other names are used in Table S3.

8.2. In line99, it is stated that the final PFS libraries are constructed as: 5'-P7-SP1-T7promoter-SP2-TerB-P7. It is not explained what SP1 and SP2 refer to, probably sequencing primer 1 and 2? Additionally, the 5' adapter is probably P5 instead of P7, the fixed sequence after the promoter is missing and the target withPFS's is missing. If understood correctly, the PFS libraries look like: 5'P5-SP1'-T7 promoter-fixed sequence-PFS-Target-PFS-SP2-TerB-P7' 3'.

8.3 In line 146, the reader is referred to a GitHub repository for the files containing information on the microscope stage designs. However, the files cannot be found in this repository.

October 19, 2023

Please find attached our revised manuscript entitled “Massively Parallel Profiling of RNA-targeting CRISPR-Cas13d” (NCOMMS-23-16199A). We thank the three reviewers for their insightful and constructive feedback. Our comprehensive point-by-point response includes the following major additions: (1) new experiments with RfxCas13d generalize our conclusions beyond EsCas13d; (2) additional binding experiments confirm that proximal mismatches reduce target RNA cleavage, whereas distal mismatches reduce binding affinity; (3) structural analysis of sub-optimal RNA structures reinforce all the main conclusions regarding the sensitivity of Cas13d binding of structured RNA targets. In addition, we’ve clarified the text and included additional discussion points, as suggested by all three referees. In sum, we have addressed all reviewers’ concerns and have strengthened the main takeaways from this study.

Reviewer #1:

Summary

This manuscript by Kuo et al. repurposes the chips used in Illumina sequencing for high throughput biophysical characterization of Cas13d binding on many RNA targets with known sequences. These sequencing chips contain clusters of DNA molecules of varying DNA sequences, with each cluster containing many DNA molecules of a particular sequence. These clusters are then subjected to in situ transcription which is then purposefully stalled by a Tus protein (a known RNA polymerase staller) bound to the flanking *terB* sequence. This stall not only stalls the transcription but causes the transcribed RNA to be tethered to the stalled complex. Thus, each DNA in the cluster has a dangling transcribed RNA whose transcription was deliberately stalled by the Tus. Therefore, you get a stalled RNA molecule from each DNA molecule in the cluster, thereby giving multiple RNA molecules (of known

sequence) for each cluster. Fluorescently tagged RNA binding proteins (like Cas13d) can then be added (& later removed) to this chip containing clusters of different RNA molecules. With the addition and removal of this fluorescently tagged protein, the changes in fluorescent signal over these clusters report on the biophysics of interactions between the Cas13d and RNA (of different known sequences). Authors call this pipeline CHAMP. These kinds of pipelines are not new and have been reported earlier including for CRISPR systems. The pioneers of these pipelines include Finkelstein group itself (authors of this manuscript) and the Greenleaf group, amongst others. These types of pipelines are a powerful method for high throughput profiling of the impact of nucleic acid sequences on the protein-nucleic acid interactions. I think they are even more helpful for the RNA because these sequences guide RNA's secondary structures. Therefore, you can profile not only the sequence dependence but also secondary structure dependence on the protein-nucleic acid interactions.

The experiments reported in the manuscript are expertly performed and reasonably clearly communicated. The number of experiments and different types of guide RNA and combinations in its sequence tested, ranging from those in protospacer flanking sequence (PFS) to those in PFS-distal and PFS-proximal region, are pretty high underlying the strength of the observations. The authors then combined this extensive experimental data with modeling that would better help understand the underlying structure of the comprehensive experimental data. So, I commend the authors for their hard work and thorough investigation. I am quite positive about this manuscript's acceptance for publication in Nature Communications. But there are some comments (one major and a lot of minor ones) whose addressal will improve the interpretability and communication of their results.

Comments

Major comment #1.1: While authors report controls to show that results from CHAMP can be recapitulated on other biophysical assays like the BLI. But one of my concerns is if all RNA molecules in a cluster are transcribed to the same length. What if the RNAP unloads

before it is stalled by Tus. I am guessing such premature unloading of RNAP will cause such RNA to detach from the DNA, allowing for a new round of transcription to occur. Therefore, only RNA molecules whose transcription reaches till Tus will remain tethered to the DNA molecule. Perhaps authors can explain this part in their methods section.

We thank the reviewer for this important point. Indeed, T7 RNAP typically undergoes abortive initiation. A recent single molecule FRET study showed that prematurely stalled T7 RNAP undergoes recycling or exchange [13]. RNAP recycling involves the restart of the stalled enzyme, whereas RNAP exchange refers to the dissociation of the stalled RNAP. Then another RNAP binds to begin a new round of RNA synthesis. In both scenarios, the transcript would indeed be generated after T7 RNAP is stalled by Tus. We include this discussion in the text:

The revised manuscript includes (page 4-5):

Tus is then added to the chip to stall T7 RNAP. In vitro transcription (IVT) and subsequent stalling of T7 RNAP tethers the transcript to its DNA template. Polymerases that stall prematurely can undergo recycling or exchange with an active enzyme [13]. In both scenarios, the transcript is generated after a transcribing RNAP is stalled by Tus.

Major comment #1.2: The second concern is the varying degrees of transcription efficiency for different sequences. All authors' analyses assume that roughly the same number of RNA molecules will be in each cluster. But I am guessing transcription would have a sequence dependence, thereby likely giving different numbers of RNA molecules per cluster, which would then affect the strength of the fluorescent signal for each cluster. Do authors do additional controls (other than BLI controls) to account for this effect?

The number of RNA molecules in a given cluster will depend on two parameters: (1) the number of DNA molecules in this cluster; and (2) in vitro transcription (IVT) efficiency. Illumina reports 500-1000 DNAs in a typical MiSeq cluster after bridge amplification. Because clustering is performed before RNA-CHAMP experiments and is dependent on the surface

density, length, and composition of the DNA library, we cannot control this parameter. However, we average over this variation. As the reviewer points out, another source of RNA variation can arise during IVT. To address this question, we performed an experiment where the transcribed clusters were hybridized with a complementary fluorescent DNA strand after IVT. We can then compare the relative distribution of fluorescent intensities as a proxy for the number of RNA molecules across our library. This is summarized in Figure R1. The DNA oligo that is hybridized to transcribed RNA shows a tight distribution of fluorescence intensities (Fig. R1A). We also see a weak correlation between the predicted structure and mean free energy (MFE) of the RNA library and the transcribed RNA signal (Fig. R1B, C). To control for the minor variation in RNA intensities, we fit the resulting binding curves without limiting I_{max} which may improve the accuracy of the apparent K_d between library sequences (Fig. R1D).

Figure R1: In vitro transcription varies minimally between sequence members in the PFS library. (A) Mean intensities of sequences in the PFS library were tightly clustered around $1.4-1.6 \times 10^5$ arbitrary units (AU). (B) The number of intramolecular base pairs is weakly correlated with fluorescence intensity. Spearman's $\rho = -0.22$. (C) The minimum free energy (MFE) is weakly correlated with fluorescence intensity. Spearman's $\rho = 0.61$. (D) Representative examples of four distinct sequences from the PFS library and their binding by increasing concentrations of Cas13d. Lines are the fits to a Hill equation without cooperatively. Blue line: 5'PFS-GUU, 3'PFS-UGC. Orange line: 5'PFS-GUU, 3'PFS-UCG. Green line: 5'PFS-GUA, 3'PFS-UUA. Black line: Non-target.

Comment #2 : While the manuscript is clearly written, I think it is a bit too heavy on Biophysics and can be simplified further to increase its reach to broader audience. A non-biophysical audience may find it difficult to understand the whole manuscript fully. For

this, I suggest the following:

Comment #2.1 : A single sentence in the main text explaining the range of values ΔABA takes and what these values mean for Cas13d and RNA binding.

The revised text includes this clarifying sentence on page 6:

The ΔABA reports the relative change in dCas13d binding affinity of every library member relative to a reference (matched target) sequence.

Comment #2.2: Figure 1B and 1G can be moved to SI figures to increase the clarity of main text figures. These two panels are not needed for main text figures.

We believe that these figures are crucial for the understanding of the first figure and subsequent experiments. Figure 1B describes the set of RNA sequences that will be probed in all subsequent experiments, along with the NGS coverage for these sequences. This is critical for understanding all downstream experiments and should be in the main text. Since RNA-CHAMP expands prior NGS-based biophysical methods, and since it is the first time such a method is applied to RNA-binding CRISPR proteins, we sought to confirm that our data correlates with more transitional measurements of binding affinity. This is indeed the case and is also captured in Figure 1G. The excellent agreement between RNA-CHAMP and established approaches (BLI) is an important validation of our method and should thus be part of Figure 1.

To clarify this point, the revised text reads (page 6):

We sequenced both RNA libraries to ensure $>\sim 10-100$ DNA clusters for all library members (**Fig. 1B**, right). ...We conclude that the massively parallel RNA-CHAMP platform can quantitatively profile protein-RNA interactions.

Comment #2 - 3: The color scheme should be improved to avoid confusion. E.g., Cyan is used for term sequence as well as target sequence in Figure 1B. Orange color is used for PFS in Figure 1B and for fiducial circles as well. These minor color inconsistencies reduce the

ease of perusing the figures, so I suggest authors develop a better color scheme that would be consistent and non-confusing throughout the manuscript.

The revised figure incorporates these suggestions and is reproduced below. We change the color of Tus and the variable region in order to avoid confusion with Fig. 1B target region and Fig. 1C T7 promoter scramble clusters. We also added a brown segment to indicate buffer sequences.

Figure 1A: RNA-CHAMP workflow. DNA is regenerated on the surface of a sequenced MiSeq chip and is transcribed with T7 RNA polymerase (RNAP). Tus retains T7 RNAP and the associated transcript on the DNA. Fluorescent dCas13d is incubated in the chip and the chip surface is imaged. The variable DNA region (blue) is flanked by a fixed sequence to maintain the same context (gray). The Tus binding site is labeled in orange.

Comment #2 - 4: Title of Cas13d requires a partially unstructured target RNA should be better titled as Cas13d has no strong PFS require but requires a partially unstructured target RNA in distal region.

We followed the overall suggestion while keeping the subtitle concise. The revised subtitle is:

Cas13d requires a partially unstructured target RNA in the distal region

Comment #2 - 5: “This preference, however, only partially explains the wide range of Δ ABAs for these datasets”. I did not fully understand the logic of this statement in main text. Can you please explain why this only partially explains? It is clearer from the subsequent analysis, but as a standalone statement, I did not understand how the sequence only partially explains the wide range of Δ ABA.

Here, we are drawing attention to the conclusion that the weak PFS preference seen in our binding dataset can't explain the broad range of observed binding affinities. The revised sentence reads (page 7):

This weak PFS preference, however, doesn't explain the broad range of Δ ABAs that we measured across the library of matched target RNA sequences.

Comment #2 - 6: "no statically significant" should be no statistically significant. The revised text is clarified below (page 7):

In contrast, we did not see any relationship between the number of intramolecular base pairs and the Δ ABA in the proximal region (**Fig. 2E, S3C**).

Comment #2 - 7: Title of Cas13d is sensitive to mismatches in the distal region of the target RNA is better titled as Cas13d binding is sensitive to mismatches in the distal region of the target RNA, especially ones that form secondary structures there.

We tried to convey the same message with a more concise sub-heading:

Cas13d binding is sensitive to mismatches in the distal region of the target RNA

Comment #3: Lack of sufficient discussion on the following significant aspects of Cas13d:
Comment #3.1: How do the preferences observed for Cas13d's binding and cleavage relate to its structure?

We include the structural paragraph in the Discussion (page 13):

Here, we show that Cas13d binding to the target RNA is penalized when the distal region is structured or is mismatched relative to the crRNA. Target RNAs with the distal region occluded by intramolecular base pairing show significant binding defects. Structures of the binary RspCas13d- and EsCas13d-crRNA complexes reveal a solvent-exposed spacer region in positions 4-8 and positions 14-20 relative to the crRNA [16, 12]. Based on the large effects of intramolecular

base pairing in the distal position (residues 14-20), we hypothesize that Cas13d initiates target recognition in this distal region (Fig. 6). Structure of ternary EsCas13d complex suggests that the helical-1 domain has the largest conformational shift compared to other subdomains. Helical-1 domain residues K376, N377, G379, K443, and Y447 are centered around the proximal region of the crRNA (positions 3-6). Mutating residues K376, K443, and Y447 to alanine fails to activate the HEPN domains [16]. Disruption of the protein-RNA interface by either a mismatched base pair or helical-1 amino acid mutations inactivates the nuclease. This indicates that the interaction of the proximal crRNA region and the helical-1 domain is critical for nuclease activation. Further kinetically resolved structural studies will be required to elucidate the mechanisms of target recognition, RNA duplex propagation, and HEPN nuclease activation.

Comment #3.2: Most CRISPR enzymes like Cas9 strongly depend on PAM-proximal region for binding and the PAM-distal region for cleavage, where this dependence is reversed for Cas13d; can you speculate about this reversal? Is it purely based on its structure? Or is there an underlying biological significance for this unusual (from other CRISPR enzymes) dependence?

To our knowledge, DNA-targeting Class I and Class II effector complexes—including Cas9—initiate target binding from the PAM-proximal side. This is because the DNA duplex begins to unwind from the PAM-proximal side. Thus, the R-loop propagates from the PAM-proximal side too. In contrast, Cas13a-d, bind RNA which is nominally single-stranded but also folded into a higher order secondary and tertiary structure. Cas13d also doesn't exhibit a strong PFS. We speculate that Cas13d preferentially initiates target binding at exposed target RNA nucleotides. As we discuss in the manuscript, the distal region of the target RNA may be a preferred entry point for target binding because it is solvent-exposed in all published Cas13d structures. In sum, the fact that Cas13d binds a (nominally) ssRNA target, and structural differences between type II/V (i.e., Cas9/Cas12a families) and type VI (Cas13 family) enzymes may result in the divergence between DNA- and RNA-targeting class 2 nucleases. We addressed this comment and revised the manuscript in relation to comment #3.1.

Comment #3.3: Maybe authors can more strongly emphasize that these kinds of high throughput profiling for sequence dependence are even more critical for RNA because they help us understand the importance of both nucleotide identities and secondary structures they create.

We emphasize the importance of considering RNA structures throughout the abstract and manuscript. We also added the following text in the Discussion (page 14):

More broadly, quantitative studies of RNA-binding CRISPR enzymes must consider the impact of RNA structure on target binding and nucleolytic activity.

Comment #3.4: CRISPR enzymes' collateral nuclease activities have been repurposed to amplify the detection of specific nucleic acid sequences, thereby helping create virus diagnostic tools. But I see that these tests have not taken off. Most of the tests I see are still PCR based. Did these tests not take off? because of {a} Lack of amplification and thus lack of signal {b} Ease of use and applicability. {b} Lack of sensitivity, which authors have shown can be improved by knowledge about the specificity of CRISPR enzymes like Cas13d. Perhaps authors can talk a little more about the future of CRISPR-based nucleic acid diagnostic tests.

CRISPR-based diagnostics are a rapidly emerging field. The first CRISPR-based diagnostic was granted emergency use authorization in May 2020, just three years ago. Work on these diagnostics platforms was supercharged by the pandemic but has leveled off as the market became saturated with free, established lateral flow tests. Nonetheless, multiple companies and academic labs are continuing to pursue CRISPR diagnostics. We direct the reviewer to an excellent summary of the state of the field [7].

As a new field, CRISPR diagnostics have to overcome multiple challenges. One technical limitation is the need for pre-amplification using recombinase polymerase amplification (RPA) or similar approaches when detecting fM target concentrations. Additional enzyme discovery and engineering can partially alleviate the need for pre-amplification. But perhaps

a bigger challenge is the business case for developing these diagnostics for at-home or point-of-care use. Clinics in the developed world have established (q)PCR workflows for most pathogens. At-home diagnostics have a very small market in the US (outside of pregnancy strips). Once production scaled up, US consumers shied away from *free and widely available* COVID tests due to a complex mixture of societal and cultural reasons (e.g., test results weren't actionable or required follow-up clinic visits anyway). We speculate that business development considerations are also limiting the market penetrance of CRISPR diagnostics. We also note that low resource settings are another important use case for these tests, especially when lab technicians and PCR equipment are limited. We believe that this discussion is beyond the scope of our study, but we have added references that summarize these points in the manuscript.

Reviewer #2:

Summary

This manuscript introduces a highly innovative method (RNA-CHAMP) to measure the binding affinity of Cas13 enzymes over many RNA targets. Beyond technology development, several important and innovative insights are also discovered using this approach, including (1) Cas13d does not have PFS preferences, (2) Cas13d binding is sensitive to mismatches mainly from distal regions; (3) target RNA base pairing predicts binding affinity; and (4) proximal mismatches do not affect binding but suppresses nuclease activity. These discoveries are uniquely made available with the proposed RNA-CHAMP technology.

Comments

Comment #1: authors use EsCas13d and its dead version (dCas13d), which is different from the widely used RfxCas13d (or CasRx). I will recommend performing RNA-CHAMP

experiments on RfxCas13d. The purpose is not to repeat every experiment in RfxCas13d, but rather to evaluate whether the major conclusions also apply to RfxCas13d, which is used much more frequently than EsCas13d.

Per the reviewer’s suggestion, we cloned and purified RfxCas13d and the nuclease-dead dRfxCas13d. To test whether our findings are specific to EsCas13d, we assayed the binding sensitivity (with dRfxCas13d) and cleavage sensitivity to proximal and distal mismatches between the crRNA and target RNAs (with RfxCas13d). These results are summarized in the Supplemental Figure 8 and reproduced below.

The manuscript was updated with RfxCas13d data (page 11):

Next, we assayed key aspects of our mechanistic insights with *Ruminococcus flavefaciens* Cas13d (RfxCas13d), as this enzyme is widely used for RNA knock-down and engineering applications [14, 4, 3]. RfxCas13d binding showed a marked sensitivity to RNA structure in the distal end of the target RNA (Fig. S8A), as measured via BLI. This binding sensitivity was strongly correlated between EsCas13d and RfxCas13d (Pearson $r = 0.93$), indicating a similar target recognition mechanism (Fig. S8B). As with EsCas13d, proximal mismatches C2G, C4A, and C7A showed very high binding affinities, but compromised cleavage (Fig. S8C, D). Taken together, we conclude that RfxCas13d and EsCas13d both penalize binding to target RNAs with distal mismatches and structures, and both exhibit a proximal cleavage sensitivity region.

Supplemental Figure 8, Binding and cleavage sensitivity of RfxCas13d (A) Relative DNA binding affinities for nuclease-dead RfxCas13d, as measured by biolayer interferometry (BLI). Target RNA sequences harbor the indicated single mismatches relative to the matched target RNA. Binding affinities are normalized to the matched target. Error bars: S.D. of three replicates. Blue: proximal mismatches; red: distal mismatches. (B) Correlation of ΔABAs of EsCas13d and RfxCas13d. Pearson $r = 0.93$, p -value < 0.001 . (C) Cleavage rates for the indicated mismatched targets, defined as the slope of the first 20 minutes of the fluorescent reporter assay (see Methods). (D) Correlation of binding affinity and cleavage rates for RfxCas13. Pearson $r = 0.74$, p -value < 0.05 . When the three proximal mismatches—C2G, C4A, and C7A—are excluded, Pearson $r = 0.95$ (p -value < 0.05).

Comment #2: Figure 2 B,C and Figure 3 D: most of the target sequence (sequence between 5' PFS and 3'PFS) form base pairs. An important finding is that the binding affinity is reversely correlated with the number of mismatches (Fig. 2E). Two associated questions:

Comment #2.1: is the RNA structure determined by computational predictions? How reliable is the predictions; i.e., are there multiple possible secondary structural predictions with similar MFE?

We reported the lowest mean free energy (MFE) RNA structure, as predicted by ViennaRNA [25]. This is a well-established RNA secondary structure prediction package and has been benchmarked extensively. Newer packages that use both conventional approaches and deep learning models have also been published [11, 17], although the deep learning approaches may not generalize across different RNA families [5].

We also compared the base pairing propensity of all suboptimal structures that have a free energy within 1 kcal/mol of the MFE. For example, for an RNA with a predicted

MFE of -14.3 kcal/mol, we compare the average number of intramolecular base pairs of all suboptimal RNAs with an MFE of -14.3 to -13.3 kcal/mol. This analysis showed a significant correlation of intramolecular base pairing and binding affinity in the distal region in both targets but not in the proximal region (**Fig. R2**). In short, this analysis confirms our main conclusions.

Figure R2: Correlation of intramolecular base pairing and Cas13d binding affinity. (A) Scatter plot of the average number of intramolecular base pairs and normalized ΔABA for target #1. Left: distal positions 12-22 (Pearson $r = -0.25$, p -value < 0.0001). Right: proximal positions 1-11 (Pearson $r = 0.0067$, p -value = 0.80) (B) Scatter plot of the average number of intramolecular base pairing and normalized ΔABA for target #2. Left: distal positions 12-22 (Pearson $r = -0.35$, p -value < 0.0001). Right: proximal positions 1-11 (Pearson $r = 0.0080$, p -value = 0.72)

Comment #2.2: By looking at the examples in Figure 2B and C, it appears the maximum length of consecutive unpaired bases is also critical, as a long consecutive unpaired base appear to have stronger binding (e.g., ACCAU in 5'PFS: GUA in Fig. 2B; and AUAGAGA in 3' PFS GCU in Fig. 2C). Would that feature (maximum length of consecutive unpaired bases) associated with delta ABA value?

Per the reviewer's suggestion, we analyze the effect of the maximum length of consecutive unpaired bases on binding affinity. We grouped target RNA sequences based on the number of consecutive exposed nucleotides on either the distal or proximal side of crRNA (**Fig. R3**). We analyzed up to four consecutive exposed nucleotides because targets #1 and #2 both had few sequences with > 4 consecutive exposed sequences. For target #1, we observed that the distal region lacking 2-4 consecutive exposed nucleotides had a lower binding affinity. The effect was not observed in the proximal region. In target #2, we also observed that

sequences lacking four consecutive exposed nucleotides in the distal region exhibit lower binding affinity. If four consecutive exposed nucleotides were in the proximal region rather than distal, it also showed stronger binding affinity. Overall, we found that sequences without consecutive unpaired bases in the distal region have weaker Cas13d binding affinity.

Figure R3: Consecutive stretches of exposed nucleotides can modulate Cas13d binding affinity. (A) Schematic of the predicted RNA structure with four consecutive exposed nucleotides in the distal region. (B) RNAs in target #1 and (C) target #2 PFS libraries were grouped by numbers of consecutive exposed nucleotides in either distal or proximal regions. (**:p-value < 0.01, ***:p-value < 0.001)

Comment #3: The main conclusion of Figure 5 A-D is that proximal mismatches do not affect binding affinity but strongly suppresses nuclease activity. Therefore, crRNA2 should have similar binding affinity but weaker cleavage in D905N than Wuhan strain. In contrast, crRNA1, whose mismatch is on the distal region at D905N, should have both weaker binding affinity and cleavage in D905N than Wuhan. Authors only show crRNA1/2 have weaker cleavage in D905N than Wuhan, but no binding comparison is shown, which will be key to confirm the conclusions in Figure 5.

Per the reviewer's suggestion, we purified the RNP of crRNA-1 and -2, and then measured their binding affinity via BLI. These results are reproduced below (page 12).

Next, we measured the binding affinity of Cas13d with crRNA-1 and crRNA-2 to both the Wuhan and Delta variants using BLI. As expected, crRNA-1 RNPs only had a weak affinity for the D950N RNA (Fig. S8E, F). In contrast, crRNA-2 RNPs had a comparable binding affinity for both target RNAs (Fig. S8E, F).

Supplemental Figure 8: Single-nucleotide changes diminish target RNA binding. (E) BLI curves of crRNA-1 and crRNA-2 with MT and D950N target RNAs. Colored lines are a global fit to three concentrations. Gray lines: measured BLI binding curves. (F) The apparent binding affinity is computed from a global fit to the data.

Minor comments

Minor comment #1: If applying the model (Figure 4) to the guides in Figure 5, do the prediction results match the experimental results?

The D950N mutation in the SARS-CoV-2 Delta strain is caused by a G→A SNP in the spike protein. We used the Wuhan sequence as our matched target and ran model VI on the Delta sequence (Fig. 4). The inputs for the model are the mismatched position compared to the matched target (Wuhan in this case), ViennaRNA-predicted intramolecular basepair positions, and the minimum free energy (MFE). Table R1 shows the predicted binding affinities when the G→A SNP is positioned at all 22 possible sites along the 22-nt spacer. Based on this analysis, we placed this SNP at position 17 since it has the lowest normalized Δ ABA (0.21).

Wuhan D950N target sequence	Predicted normalized Δ ABA
AGUGCACUUGGAAAACUUCAAG G	0.48
GUGCACUUGGAAAACUUCAAG A	0.40
UGCACUUGGAAAACUUCAAG AU	0.33
GCACUUGGAAAACUUCAAG AUG	0.43
CACUUGGAAAACUUCAAG AUGU	0.39
ACUUGGAAAACUUCAAG AUGUG	0.44
CUUGGAAAACUUCAAG AUGUGG	0.38
UUGGAAAACUUCAAG AUGUGGU	0.26
UGGAAAACUUCAAG AUGUGGUC	0.37
GGAAAACUUCAAG AUGUGGUCA	0.45
GAAAACUUCAAG AUGUGGUCAA	0.28
AAAACUUCAAG AUGUGGUCAAC	0.52
AAACUUCAAG AUGUGGUCAACC	0.37
AACUUCAAG AUGUGGUCAACCA	0.27
ACUUCAAG AUGUGGUCAACCAA	0.30
CUUCAAG AUGUGGUCAACCAAA	0.49
UCAAG AUGUGGUCAACCAAAA	0.21
UCAAG AUGUGGUCAACCAAAA	0.40
CAAG AUGUGGUCAACCAAAA	0.23
AAG AUGUGGUCAACCAAAA	0.64
AG AUGUGGUCAACCAAAA	0.37
GA AUGUGGUCAACCAAAA	0.35

Table R1: Relative binding affinities for the indicated crRNA, as calculated by model VI. The G→A SNP is underlined.

Reviewer #3

Summary

Kuo et al has advanced their previous high throughput fluorescence method, CHAMP, for studying RNA studies. One notable novelty of the study is the decoupling of binding and cleavage, which distinguishes it from previous investigations. However, to enhance this aspect, the authors should provide further clarification on the rationale behind certain choices and elaborate on the obtained results. For example, it would be advantageous to expand the high-throughput analysis of cleavage beyond the current limited set of nine sequences.

Furthermore, it is important to address certain points to improve the overall understanding of the study. Specifically, the authors should explain the reasons behind a significant

number of library members yielding results below the detection threshold. Additionally, clarification is needed regarding the fixed sequences adjacent to the target that introduce secondary structure. Consideration should be given to including a control group without strong secondary structures to serve as a baseline for comparison.

Overall, the manuscript should be rewritten to acknowledge the existing literature properly and avoid giving the impression that certain characteristics of Cas13d are being revealed for the first time. While the study does contribute novel findings, it would be beneficial to strengthen certain aspects by providing clearer explanations, additional information, and expanding the scope of experimentation.

Major comments

Comment #1: Concerning the introduction

Comment #1.1 :The current introduction reads as if RNA-CHAMP is a completely new method. I ask the authors to mention RNA-MaP and their own CHAMP in the introduction (or first results section) instead of in the discussion.

We revised the introduction and included the following section (page 4):

Here, we describe RNA-CHAMP (chip-hybridized association-mapping platform) for massively parallel profiling of RNA-protein interactions on a conventional microscope and the nearly ubiquitous Illumina MiSeq chips. Our approach differs from prior high-throughput methods [23, 24] that repurpose the obsolete Illumina Genome Analyzer IIx instruments and require custom hardware modifications [21].

Comment #1.2: The introduction does not give a correct picture of what is and what is not already known about EsCas13d. The current text explains that different Cas13-family members have different or no PFS requirements and refers to one study on the effect of mismatches on Cas13d activity. However, some very relevant previous findings specific to Cas13d are missing. For example, the already existing evidence and structural explanation

for the lack of a PFS requirement for Cas13d (Yan et al. Mol Cell 2018, Konermann et al. Cell 2018, Zhang et al. Cell 2018) are not mentioned. Additionally, in previous studies there were already observations of a negative correlation between secondary structure of the target RNA and Cas13d efficiency (e.g. Yan et al. MolCell 2018). This is mentioned later (line 320) but it should be informed in the introduction.

We revised the sentence and included EsCas13d. We also include a sentence describing the observation of secondary structure on Cas13d targeting efficiency in the introduction (page 3).

However, LwaCas13a, PspCas13b, EsCas13d, RfxCas13d (CasRx), and Cas13X.1 may not require any PFS at all[22, 15, 14, 8, 16, 18, 19]. ...Additionally, prior reports suggested that the secondary structure of the target is negatively correlated with Cas13d targeting efficiency[15, 10, 6].

Comment #2: Concerning the section “Cas13d requires a partially unstructured target RNA”: In line 148, the authors describe that they randomized three nucleotides on both the 3’ and 5’ end of a perfectly matched target and that they measured the Δ ABA for 1457 different combinations. The other sequences, more than half of the in total 4096 sequences, were below the detection threshold. It would be very informative to know whether these sequences have something in common (e.g. a very strong secondary structure) that explains why they could not be measured or whether there is another explanation for such a large part of the library, with perfectly matched target sequence, being below the detection threshold.

Figure R4 highlights our data fitting and analysis process. For example, **Fig. R4A** shows two sequences with good fits and three sequences with low-quality fits that are omitted from the final analysis. The fits in blue and orange are examples of sequences that fit well and were thus included in the downstream analysis. The most common failure mode is a very small change in the fluorescent intensity at the highest Cas13d concentrations (green and red fit in (**Fig. R4A**)). The red line shows a sequence with a low apparent K_d . The green line is

typical of sequences with weak but measurable binding that nonetheless didn't saturate up to the experimental maximum protein concentration of 128 nM. We hypothesize that the low binding intensity is mainly caused by the RNA structure in those sequences; cluster-specific effects are averaged out over the 10+ clusters that make up every sequence in the library. We successfully collected measurements for 2533 and 2083 out of 4096 possible sequences in two replicate PFS library experiments (**Fig. R4B**). Thus, we analyzed sequences that had a good fit in both replicates and within the experimental range of 1-128 nM for the apparent K_d .

Next, we compared the predicted structure of sequences with measurable K_d s and the sequences that were below our detection threshold. We found that measurable sequences have overall higher MFE than sequences that resulted in poor fits (**Fig. R4C**). The number of predicted intramolecular base pairs in the measurable sequences is lower in the distal region and higher in the proximal region. We conclude that sequences with high predicted intramolecular base pairs show low binding affinity in the RNA-CHAMP assays.

Figure R4: Sequences that are below the RNA-CHAMP detection threshold are highly structured. (A) Curve fitting of highlighted sequences. Blue and orange lines are examples of sequences with a high-quality fit. Red and green lines are examples of sequences below our detection threshold. Black line: a scrambled sequence that serves as a negative control. (B) Scatter plot of ΔABA of replicate experiments. Dashed lines indicate the detection limit. (C) Structured RNAs are enriched in sequences with poor fits. (***:p-value < 0.001)

Comment #3: Concerning the section “Cas13d is sensitive to mismatches in the distal region of the target RNA

Comment #3.1: In line 190 the authors state that the binding trends were broadly the same across the two libraries. Comparing Fig. 3A and Fig. S4A, I agree that broadly there is a similar trend but there is a difference in where the mismatch-sensitive region starts as also acknowledged by the authors in line 199. Would this be due to the stronger secondary structure and thus weaker binding of target #3 as mentioned later in line 217? What is the expectation for how target-dependent the position of the mismatch-sensitive region is?

We think that the slightly different sensitivity region is due to the combination of the target sequence and secondary structure. As the reviewer noted, the secondary structures of matched targets #1 (Fig. 2B) and #3 (Fig. S4C) are drastically different. Target #3 forms a strong secondary structure that weakens the binding affinity. Therefore, we have lower dynamic ranges in target #3 sequences. A second effect may be the preference for GC-rich sequences in the distal region [1, 6, 10]. Targets #1 and #3 have 55 % and 27 % GC content respectively. This can also contribute to the slightly wider sensitivity region.

Comment #3.2: From the results and structural predictions presented in Figure 2 and 3, it becomes clear that the fixed sequences next to the target and PFSs form secondary structures with the target. What is the motivation for placing these sequences within the transcribed RNA and not after the *terB* and before the T7 promoter site, respectively? The fixed sequence at the 3' side of the target is a sequencing primer but what is the sequence on the 5' side of the target?

As reviewer #3 stated, the sequences we designed in the 3' side of the target is Illumina sequencing primer 2 (SP2). We put *TerB* after the SP2 to (1) provide a buffer sequence that allows the stalled transcript to exit the stalled T7 RNAP exit channel; (2) accommodate future libraries with changes in the sequence length, such as long insertions and deletions; (3) reduce unnecessary reads of constant regions and maximize paired-end coverage. The purpose of the constant sequence on the 5' side of the target is to even out varied transcription activities from the diversity of our library members. T7 RNAP transcription is largely affected by the first few bases [9] which would strongly affect the transcription level

if the library were placed directly after the promoter. Please see our response to reviewer #1's major comment 1.1 for a further discussion on how we assayed the effects of variable transcription across the RNA library.

Comment #3.3: Relatedly, to decouple the secondary structure and sequence effect, a control experiment should be performed where the flanking fixed sequence does not form secondary structures with the target.

We performed this control experiment with a 22 nt target RNA flanked with U_9 :

5'-UUUUUUUUUCCAUAAGAGAGGUUAUCCGCUCAUUUUUUUUU-3'

This sequence only allows a secondary structure to form within the target region. Figure R5 compares the binding affinity of this sequence to the original target. The original sequence has a dissociation constant (K_d) of 3.1 ± 0.018 nM, whereas the flanking-U sequence has a K_d of 1.6 ± 0.024 nM, as would be expected for a slightly less structured target.

Figure R5: BLI analysis of MT without flanking structures. Colored lines are global fit to 100 nM, 50 nM, and 25 nM (top to bottom). Gray lines are raw BLI curves. The dashed line represents the time point of measuring the off-rate.

Comment #3.4: In line 227 the authors state “However, insertions and deletions at the distal side of the target RNA on both targets were not tolerated (Fig. S5, S6)”. However, for the target in Fig. S5, many of the insertions decrease ΔABA only slightly. For the target in Fig. S6 almost all insertions/deletions (also in the proximal region) abolish detectable binding, but as was already mentioned before this target is highly structured and thus very

weakly bound by Cas13d. Please comment.

We outlined the main difference between these targets in Comment #3.1. Differential indel sensitivity for targets #1 and #3 is mainly due to the lower dynamic range in the highly structured target # 3. Target # 3 has generally lower binding affinity as compared to target #1, so indels and substitutions are more likely to drop the binding affinity below RNA-CHAMP’s deletion limit.

Comment #3.5: To strengthen their claim about the effect of secondary structure, could the authors consider providing a plot with Δ ABA versus the number of intramolecular base pairs in the target instead of picking out two examples?

Per the reviewer’s suggestion, we analyzed the effect of secondary structure on target #1 (Fig. R6). We found that the intramolecular base pairing has a strong negative correlation in the distal region in target #1. We did not analyze target #3 because it is very structured; only a few sequences reduce intramolecular base pairing.

Figure R6: Intramolecular base pairing reduced the Cas13d binding affinity. RNA structural analysis in target library #1. The predicted structure of the target library was grouped by the number of intramolecular base pairs. (**: p-value < 0.01)

Comment #4: Concerning the biophysical model: How does the performance of the presented biophysical model compare to DeepCas13 (Cheng et al. Nat Commun 2023)? DeepCas13 predicts on-target activity while the biophysical model presented by the authors is about binding, but it would be very interesting to see the differences and whether they can

be explained by the insights the authors gained by decoupling binding and cleavage activity.

DeepCas13 is a model for predicting on-target efficiency and specificity in mammalian cells [1]. This model was trained on data from 10,830 crRNAs targeting 192 protein-coding genes and 234 lncRNAs in a melanoma cell line. In contrast, our model is focused on predicting the relative binding affinity to a given crRNA-target sequence with certain mismatches, indels, and secondary structure changes. Our library is therefore constructed with a few crRNAs but a large pool of alterations such as mismatches, insertions, and deletions. These experiments are also conducted in vitro and quantitatively measure RNA binding, which we show in Fig. 5 can be very different from cleavage. In short, we believe that these two models are designed to report on different aspects of Cas13d biology and should not be compared directly (even a positive correlation would be difficult to interpret).

Comment #5: Concerning the cleavage data

Comment #5.1: Figure 5D, what is the Pearson correlation coefficient? With 9 sequences, of which 3 are deemed outliers (more than half of the proximal mismatch sequences), it is hard to draw a conclusion about general rules for the effect of mismatches on Cas13d cleavage. And, as their conclusion contradicts data from a large-scale Cas13d screen in mammalian cells by Wessels et al. Nat.Biotech. 2020, the authors should compare their cleavage data with the cleavage data (not just binding data) from Wessels et al.

The Pearson correlation coefficient on all sequences is $r = 0.035$ (p-value = 0.92), where the correlation coefficient without those three outliers is $r = 0.78$ (p-value = 0.040). To get a comprehensive view of the proximal mismatch on Cas13d cleavage, we performed the cleavage experiment on positions 1-11 and observed a similar trend in the main figure. The cleavage activity is gradually decreased from position 11 to proximal mismatches, and Cas13d has very minimal activity with mismatches in positions 1-6.

Figure R7: Single mismatch analysis of Cas13d cleavage activity in the proximal region. (A) Raw cleavage data of all proximal sequences. (B) 20-mins slope (AU/min) of all proximal sequences

Studies from Wessels et al. [10, 2] discovered a bimodal penalty distribution in consecutive mismatch analysis. They discovered that mismatches in both positions 4-7 and positions 17-20 have reduced the cleavage activity of Cas13d. Where distal positions have a higher sensitivity to mismatch than proximal positions. In our manuscript, we described that Cas13d binding is sensitive to distal alterations and cleavage is sensitive to proximal alterations. We speculate the bimodal penalty distribution seen by Wessel et al is due to the distinct binding and cleavage sensitivity regions, as shown in our study.

Comment #5.2: The authors have chosen only 9 sequences to study cleavage. Please justify why these 9 sequences were chosen. How about the use of NucleaSeq developed in their earlier publication (Jones et al. 2021, Nature Biot)?

Cas13d has a robust *trans* RNase activity. This is activated when the RNP binds a matched target. This cleavage activity means that NucleaSeq and other pooled sequencing-based approaches would not be possible with Cas13d. Instead, we had to select a subset of sequences for single experiments. In our previous NucleaSeq paper, we avoided this problem with AsCas12a because that enzyme does not cleave double-stranded DNA efficiently *in trans*. We are planning to develop a massively parallel cleavage assay for enzymes with *trans* cleavage activity, but this work is beyond the scope of the current study.

Comment #6: Concerning the proof-of-principle for SNP detection Comment #6.1: As it is not the first time that Cas13d was used for single nucleotide variation, they should cite Qiao et al. *Biotechnol. Bioeng.* (2021).

We added this citation to the manuscript.

Comment #6.2: Both in the introduction (line 75) and in line 291 the authors state that using the insights gained from this study they can rationally design assays to detect single nucleotide polymorphisms (SNPs) in a target RNA. However, it does not become clear what the rationale is behind the design of the two used crRNAs. In line 296 the authors also mention that the two crRNAs were designed such that the SNP is positioned within a sensitive region of Cas13d activity. Can the authors pinpoint the sensitive region of Cas13d activity? With the claims that distal mismatches strongly decrease binding and cleavage and proximal mismatches inhibit cleavage, Cas13d activity seems sensitive to mismatches regardless of their position within the target RNA. Why were positions 1 and 17 chosen and would a mismatch at any position work?

The rationale for placing a mismatch at position 17 (crRNA-1) and 1 (crRNA-2) is based on the binding model (**Fig. 4**) and cleavage results (**Fig. 5**), respectively. We use model VI—our most predictive model for Cas13d binding—to predict all possible crRNAs that could target the Delta SARS-CoV-2 SNP (**R1**). This model predicts that a mismatch at position 17 will minimize the Cas13d binding affinity (normalized ABA of 0.21). crRNA-2 was designed with a mismatch at position 1 based on our cleavage assay. This assay showed that the cleavage activity is strongly compromised by mismatches in the proximal region **Figure 5A-C**. We further performed a comprehensive analysis on all proximal mismatch sequences (**Fig. R7**), and the results showed a strong cleavage inhibition in the proximal region, including position 1. Placing the mismatch in other positions may work (eg. position 11) (**Fig. R7**), but might not minimize cleavage relative to the matched target.

Comment #6.3: The proof-of-principle nicely demonstrates that two SARS-CoV-2 vari-

ants can be distinguished using Cas13d with a crRNA that overlaps with an SNP in the target RNAs. However, an alternative explanation for why this difference occurs is possible. As shown in the RNA-CHAMP assay with the PFS library, the secondary structure of the target RNA plays a large role in determining Cas13d activity. The SNP in the SARS-CoV-2 variants can affect the secondary structure. With the current data, it cannot be ruled out that even a matching guide would cleave the Delta variant less efficiently than the Wuhan variant. Therefore, a control with a matching guide for the Delta variant would help to say with more certainty why there is a difference in cleavage activity between the two variants.

To address this question, we purified Cas13d RNPs with a crRNA that matches the delta variant and performed identical cleavage experiments. The cleavage activity is comparable to the matching crRNA targeting Wuhan variant (**Fig. S8H, I**). Both sequences show an identical secondary structure (**Fig. S8G**). Therefore, we conclude that the different cleavage activities are mainly due to the effect of the mismatched base on Cas13d nuclease activation rather than RNA structure.

The supplement is revised with the additional experiment.

Supplemental Figure 8: Cas13d cleavage is comparable between Wuhan and Delta SARS-CoV-2 RNA targets. (G) A single C→A substitution differentiates the Wuhan and Delta strains (circled). Notably, this substitution is not predicted to change the target RNA structure. (H) Time traces of the fluorescent signal for each of the indicated target RNA sequences. (I) 20-mins slope (AU/min) of matched target targeting Wuhan and Delta variant.

We also added the following to the main text (page 12):

To confirm that the less efficient cleavage of the Delta variant is not due to RNA structural changes, we analyzed the predicted MFE structure. The SNP in this sequence doesn't alter the RNA structure (Fig. S8G). As expected, the cleavage rate of the crRNA that matches the Delta sequence is statistically indistinguishable from the cleavage rate of the original matched target crRNA (Fig. S8H, I). The results confirm that the SNP indeed alters the cleavage activity, and the effect is due to the position of the mismatch relative to the crRNA.

Comment #7: In the Discussion, line 325, the authors state “Cas13d requires base pairing in positions 1-6 to activate its nuclease activity.” Which data is the basis for this range?

This is primarily based on the cleavage experiments in Figures 5B, C. We also include a cleavage activity screen of proximal sequences (response to Comment #5.1) which showed a significant drop in cleavage for sequences with mismatches in positions 1-6 of the crRNA.

Minor comments

Comment #1: In Figure 1A the schematic suggests that the RNA only contains the variable region, while it also contains a fixed sequence of 24-34 nt on both sides of the target. It would be helpful for the reader if these fixed sequences are also indicated in the schematic.

The revised figure, reproduced below, includes the constant “buffer” region and an updated color scheme for clarity.

Fig. 1A: RNA-CHAMP workflow. DNA is regenerated on the surface of a sequenced MiSeq chip and is transcribed with T7 RNA polymerase (RNAP). Tus retains T7 RNAP and the associated transcript on the DNA. Fluorescent Cas13d is incubated in the chip and the chip surface is imaged. The variable DNA region (blue) is flanked by a fixed sequence to maintain the same context (gray). The Tus binding site is labeled in orange.

Comment #2: In the caption for Figure 1B, bottom, left and right are switched.

Fixed. The revised caption reads:

Fig. 1B: Top: schematic of the RNA library. The 22-nucleotide target RNA (blue) is flanked on both ends by three random nucleotides (PFS, gray) and buffer sequences (light brown). Bottom: summary of the unique DNA sequences in the synthetic library (left), and the number of clusters observed via NGS for each unique library member (right).

Comment #3: In line 99 the authors state “Importantly, Tus did not bind clusters that lacked *terB*.” However, in Figure S1A, The bar plot does not include a bar for the clusters lacking *terB*.

A subset of the DNA clusters on the MiSeq chips does not include any *TerB* sites. We include an analysis of Tus co-localizing on these DNA clusters in Figure S1A (reproduced below).

Fig. S1A: Tus binding is dependent on the *TerB* DNA sequence. Statistical analysis was performed using unpaired Student’s t-test, *** $p < 0.001$.

Comment #4: In line 158 reference 4 is missing.

We added this reference.

Comment #5: For Figure 3A, please add in the caption what the solid lines represent. Assumably the average of all mutations for each position?

The updated caption now reads:

(A) Summary of single mismatch-dependent changes in the Δ ABA for two biological replicates. Upper dashed line: matched target Δ ABA. Lower dashed line: RNA-CHAMP detection limit. Solid lines: the mean of all three substitutions.

Comment #6: For Figure 4E, please add in the caption what the bottom ‘Base pairing’ penalty is for. Actually, the authors might consider changing the term base pairing (also in the main text) to something like ‘internal base pairing’ or ‘intramolecular base pairing’ because the reader might become confused with base pairing between the target and crRNA instead of internal base pairing within the target.

We thank the reviewer for this excellent point. We have changed the term ‘base pairing’ to ‘intramolecular base pairing’ where appropriate throughout the manuscript and figures. As an example, revised figure 4E is reproduced below.

Fig. 4E: Top: weight penalty for all mismatches, insertions, and deletions. Bottom: weight penalty for intramolecular base pairs in model VI.

Comment #7: In lines 294-296 the authors state: “Here, the matched target is from the spike gene of the original(“Wuhan”) SARS-CoV-2, but harbors a single G to A substituting

(D950N in amino acid sequence) in the position 1 and 17 in the Delta VOC.” This sentence is confusing as it suggests that there are two mutations in the Delta VOC with respect to the Wuhan variant, while there is a single mutation which is placed at position 1 and 17 of crRNA-2 and crRNA-1, respectively.

The modified text reads (page 12):

Here, the matched target is from the spike gene of SARS-CoV-2. The G→A single nucleotide polymorphism (SNP) differentiates the original “Wuhan” strain and the Delta VOC. We designed two crRNAs: the first places this SNP within the binding sensitivity region (crRNA-1; position 17), and the second is in the cleavage sensitivity region (crRNA-2; position 1). Both crRNAs reduce Cas13d cleavage ~5-fold for the D950N RNA (Fig. 5F, G).

Comment #8: The methods section would benefit from a revision as it contains several inconsistencies and textual errors. Among others the following (line numbers refer to the lines in the supplementary file): Comment #8.1: In line 96 there is a reference to primers ‘JK044’ and ‘JK045’ while other names are used in Table S3.

We revised the methods section. We paid special attention to oligo and plasmid numbering.

Comment #8.2: In line 99, it is stated that the final PFS libraries are constructed as: 5'-P7-SP1-T7promoter-SP2-TerB-P7. It is not explained what SP1 and SP2 refer to, probably sequencing primer 1 and 2? Additionally, the 5' adapter is probably P5 instead of P7, the fixed sequence after the promoter is missing and the target with PFS's is missing. If understood correctly, the PFS libraries look like: 5'P5-SP1'-T7 promoter-buffer sequence-PFS-Target-PFS-SP2-TerB-P7' 3'.

We thank the reviewer for catching the P5/P7 typo, and for clarifying the jargon in this text. SP1 and SP2 refer to the standard Illumina sequencing primers. SP1 is 5'-CACTCTTTCCCTACACGACGCTCTTCCGATCT-3' and SP2 is 5'-GTGACTGGAGTTCAGACGTGTGCTCTTCCGATCT-3'. The 5'-end of the construct had a P5, not a P7 adapter.

The revised supplement reads:

Final amplified libraries were constructed as 5'-P5-SP1-buffer sequence-T7 promoter-PFS-target-PFS-SP2-TerB-P7-3'. P5 and P7 are Illumina adaptors, while SP1 and SP2 are Illumina sequencing primers.

Comment #8.3: In line 146, the reader is referred to a GitHub repository for the files containing information on the microscope stage designs. However, the files cannot be found in this repository.

The manuscript sites the GitHub repository relating to this paper. However, the microscope stage designs and CAD files are stored in a separate repository that is associated with the first CHAMP paper[20]. To streamline this, we now include the microscope stage designs in the RNA-CHAMP repository and also include a detailed README file explaining the relationship between the two repositories.

References

- [1] Xiaolong Cheng et al. “Modeling CRISPR-Cas13d on-target and off-target effects using machine learning approaches”. *Nature Communications* 14.1 (2023), p. 752.
- [2] Hans-Hermann Wessels et al. “Prediction of on-target and off-target activity of CRISPR-Cas13d guide RNAs using deep learning”. *Nature Biotechnology* (2023), pp. 1–10.
- [3] Rahul Gupta et al. “Cas13d: a new molecular scissor for transcriptome engineering”. *Frontiers in Cell and Developmental Biology* 10 (2022), p. 866800.
- [4] Martyna Kordyś, Raneet Sen, and Zbigniew Warkocki. “Applications of the versatile CRISPR-Cas13 RNA targeting system”. *Wiley Interdisciplinary Reviews: RNA* 13.3 (2022), e1694.

- [5] Marcell Szikszai, Michael Wise, Amitava Datta, Max Ward, and David H Mathews. “Deep learning models for RNA secondary structure prediction (probably) do not generalize across families”. *Bioinformatics* 38.16 (2022). Ed. by Yann Ponty, pp. 3892–3899. DOI: [10.1093/bioinformatics/btac415](https://doi.org/10.1093/bioinformatics/btac415). URL: <https://doi.org/10.1093/bioinformatics/btac415>.
- [6] Jingyi Wei et al. “Deep learning and CRISPR-Cas13d ortholog discovery for optimized RNA targeting”. *bioRxiv* (2022). DOI: [10.1101/2021.09.14.460134](https://doi.org/10.1101/2021.09.14.460134). eprint: <https://www.biorxiv.org/content/early/2022/07/08/2021.09.14.460134.full.pdf>. URL: <https://www.biorxiv.org/content/early/2022/07/08/2021.09.14.460134>.
- [7] Michael M Kaminski, Omar O Abudayyeh, Jonathan S Gootenberg, Feng Zhang, and James J Collins. “CRISPR-based diagnostics”. *Nature Biomedical Engineering* 5.7 (2021), pp. 643–656.
- [8] Chunlong Xu et al. “Programmable RNA editing with compact CRISPR–Cas13 systems from uncultivated microbes”. *Nature methods* 18.5 (2021), pp. 499–506.
- [9] Thomas Conrad, Izabela Plumbom, Maria Alcobendas, Ramon Vidal, and Sascha Sauer. “Maximizing transcription of nucleic acids with efficient T7 promoters”. *Communications Biology* 3.1 (2020), p. 439.
- [10] Hans-Hermann Wessels et al. “Massively parallel Cas13 screens reveal principles for guide RNA design”. *Nature biotechnology* 38.6 (2020), pp. 722–727.
- [11] David H. Mathews. “How to benchmark RNA secondary structure prediction accuracy”. *Methods* 162-163 (2019), pp. 60–67. DOI: [10.1016/j.ymeth.2019.04.003](https://doi.org/10.1016/j.ymeth.2019.04.003). URL: <https://doi.org/10.1016/j.ymeth.2019.04.003>.
- [12] Bo Zhang et al. “Two HEPN domains dictate CRISPR RNA maturation and target cleavage in Cas13d”. *Nature communications* 10.1 (2019), p. 2544.

- [13] Hye Ran Koh et al. “Correlating transcription initiation and conformational changes by a single-subunit RNA polymerase with near base-pair resolution”. *Molecular cell* 70.4 (2018), pp. 695–706.
- [14] Silvana Konermann et al. “Transcriptome engineering with RNA-targeting type VI-D CRISPR effectors”. *Cell* 173.3 (2018), pp. 665–676.
- [15] Winston X Yan et al. “Cas13d is a compact RNA-targeting type VI CRISPR effector positively modulated by a WYL-domain-containing accessory protein”. *Molecular cell* 70.2 (2018), pp. 327–339.
- [16] Cheng Zhang et al. “Structural basis for the RNA-guided ribonuclease activity of CRISPR-Cas13d”. *Cell* 175.1 (2018), pp. 212–223.
- [17] Yunjie Zhao, Jun Wang, Chen Zeng, and Yi Xiao. “Evaluation of RNA secondary structure prediction for both base-pairing and topology”. *Biophysics Reports* 4.3 (2018), pp. 123–132. DOI: [10.1007/s41048-018-0058-y](https://doi.org/10.1007/s41048-018-0058-y). URL: <https://doi.org/10.1007/s41048-018-0058-y>.
- [18] Omar O Abudayyeh et al. “RNA targeting with CRISPR–Cas13”. *Nature* 550.7675 (2017), pp. 280–284.
- [19] David BT Cox et al. “RNA editing with CRISPR-Cas13”. *Science* 358.6366 (2017), pp. 1019–1027.
- [20] Cheulhee Jung et al. “Massively parallel biophysical analysis of CRISPR-Cas complexes on next generation sequencing chips”. *Cell* 170.1 (2017), pp. 35–47.
- [21] Richard She et al. “Comprehensive and quantitative mapping of RNA–protein interactions across a transcribed eukaryotic genome”. *Proceedings of the National Academy of Sciences* 114.14 (2017), pp. 3619–3624.
- [22] Omar O Abudayyeh et al. “C2c2 is a single-component programmable RNA-guided RNA-targeting CRISPR effector”. *Science* 353.6299 (2016), aaf5573.

- [23] Jason D Buenrostro et al. “Quantitative analysis of RNA-protein interactions on a massively parallel array reveals biophysical and evolutionary landscapes”. *Nature biotechnology* 32.6 (2014), pp. 562–568.
- [24] Jacob M Tome et al. “Comprehensive analysis of RNA-protein interactions by high-throughput sequencing–RNA affinity profiling”. *Nature methods* 11.6 (2014), pp. 683–688.
- [25] Ronny Lorenz et al. “ViennaRNA Package 2.0”. *Algorithms for molecular biology* 6 (2011), pp. 1–14.

Reviewers' Comments:

Reviewer #1:

Remarks to the Author:

The authors have addressed all my points and concerns. The revised manuscript is of top quality, which I recommend for publication.

Reviewer #2:

Remarks to the Author:

Authors did a wonderful job in addressing the comments. As a minor point, I recommend placing Figure R2 into the supplementary figures -- these results will be important for researchers to study the effect of RNA secondary structure on Cas13 binding.

Reviewer #3:

Remarks to the Author:

This referee is generally supportive of its publication. However, this referee has one notable concern regarding the large number of sequences falling below the detection limit, particularly those with 22 nucleotides matching targets with 3 N's on both sides. It would be beneficial to address whether the observed differences in secondary structure can account for why these sequences consistently register below the detection limit.

Additionally, while this referee appreciates the authors' responses to the previous comments, it is noticed that in certain instances, they have not made corresponding adjustments in the manuscript itself. It is important to consider whether certain clarifications or changes in the text would benefit the readers even if not explicitly requested by the reviewers.

For instance, Comment #5.1 highlights that the new data with mismatches in positions 1-11 confirms the impact of proximal mismatches on reporter RNA cleavage. This referee recommends integrating this new data into Figure 5 and refraining from labeling the proximal mismatch data points as outliers, as they represent a distinct population. Emphasize the existence of two populations: 1) distal mismatches where Δ ABA correlates with cleavage, and 2) proximal mismatches where this correlation is not observed.

Furthermore, Comment #6 points to a previous publication on SNP detection, and while the authors now cite the paper, they still present the SNP detection in the two SARS-CoV-2 variants as 'a proof of principle.' It raises the question of whether such a proof of principle can still be claimed if another proof of principle has been published before. Therefore, it might be advisable to consider changing the phrasing to emphasize that the focus is on their pipeline for rationally designing the guides, rather than presenting it as a novel proof of principle.

Finally, I have a minor inquiry about Figure R4C. It is not clear if something is missing in the figure at its far right, or if it represents a single measured sequence with proximal base pairing. In light of this, I am also curious about whether the observed differences are substantial enough to account for the high number of sequences falling below the detection limit. It would be valuable to explore if there are specific types of secondary structures or other pertinent information when comparing undetected and detected sequences.

In conclusion, the manuscript shows promise but requires some minor revisions and clarifications to address the issues raised in this report.

Reviewer #1:

Summary

The authors have addressed all my points and concerns. The revised manuscript is of top quality, which I recommend for publication.

Reviewer #2:

Summary

Authors did a wonderful job in addressing the comments. As a minor point, I recommend placing Figure R2 into the supplementary figures – these results will be important for researchers to study the effect of RNA secondary structure on Cas13 binding.

We thank the reviewer for the suggestion, now Figure R2 was placed into the supplementary figure S3E, F.

Reviewer #3:

Summary

This referee is generally supportive of its publication. However, this referee has one notable concern regarding the large number of sequences falling below the detection limit, particularly those with 22 nucleotides matching targets with 3 N's on both sides. It would be beneficial to address whether the observed differences in secondary structure can account for why these sequences consistently register below the detection limit.

Additionally, while this referee appreciates the authors' responses to the previous comments, it is noticed that in certain instances, they have not made corresponding adjustments

in the manuscript itself. It is important to consider whether certain clarifications or changes in the text would benefit the readers even if not explicitly requested by the reviewers.

In conclusion, the manuscript shows promise but requires some minor revisions and clarifications to address the issues raised in this report.

Comments

Comment #1: For instance, Comment #5.1 highlights that the new data with mismatches in positions 1-11 confirms the impact of proximal mismatches on reporter RNA cleavage. This referee recommends integrating this new data into Figure 5 and refraining from labeling the proximal mismatch data points as outliers, as they represent a distinct population. Emphasize the existence of two populations: 1) distal mismatches where Δ ABA correlates with cleavage, and 2) proximal mismatches where this correlation is not observed.

Per the reviewer's suggestion, we avoid using the term "outliers."

Cleavage rates were generally correlated with ABAs, with two distinct populations (Fig. 5B-D). Proximal mismatches (i.e., C2G, C4A, and C7A) did not impact RNA binding but only weakly cleaved the reporter RNA.

We also integrated the cleavage data in **Figure 5**, reproduced below.

Figure 5

Comment #2: Furthermore, Comment #6 points to a previous publication on SNP detection, and while the authors now cite the paper, they still present the SNP detection in the two SARS-CoV-2 variants as 'a proof of principle.' It raises the question of whether such a proof of principle can still be claimed if another proof of principle has been published before. Therefore, it might be advisable to consider changing the phrasing to emphasize that the focus is on their pipeline for rationally designing the guides, rather than presenting it as a novel proof of principle.

We note that the proof of principle here refers to using our biophysical model specifically (as pointed out by the reviewer here). We rephrased this point (page 12):

As a proof of principle of our analysis pipeline, we positioned the SNP in the crRNA-target RNA duplex to differentiate between two SARS-CoV-2 variants of concern (VOC) (Fig. 5E).

and:

These results demonstrate that our analysis pipeline can be used to design Cas13d-based diagnostics that distinguish between SNPs by precisely positioning the expected mismatched positions relative to the crRNA.

Comment #3: Finally, I have a minor inquiry about Figure R4C. It is not clear if something is missing in the figure at its far right, or if it represents a single measured sequence with proximal base pairing. In light of this, I am also curious about whether the observed differences are substantial enough to account for the high number of sequences falling below the detection limit. It would be valuable to explore if there are specific types of secondary structures or other pertinent information when comparing undetected and detected sequences.

This library is skewed to eight basepairs in the proximal position, making the box plot appear incorrect (>1,000 library members with 8 intramolecular basepairs). To avoid this confusion, we replot the data as violin plots below.

Figure R4 (C) Structured RNAs are enriched in sequences with poor fits. Statistical analysis was performed using unpaired Student's t-test, *** $p < 0.001$.

We are indeed following up on some of these sequences using a combination of quantitative kinetics and structural biology. However, we believe that a detailed analysis of these effects is beyond the scope of this manuscript.